# A global spatially Contiguous Solar Induced Fluorescence (CSIF) dataset using neural networks

Yao Zhang[1], Joanna Joiner[2], Seyed Hamed Alemohammad[3], Sha Zhou[1], Pierre Gentine[1,4]

[1]Department of Earth and Environmental Engineering, Columbia University, New York, NY 10027, USA
[2]NASA Goddard Space Flight Center, Greenbelt, MD 20771, USA
[3]Radiant.Earth, Washington, D.C. 20005, USA
[4]Earth Institute, Columbia University, New York, NY 10027, USA

*Correspondence to*: Yao Zhang (zy2309@columbia.edu)

**Abstract.** Satellite-retrieved Solar Induced Chlorophyll Fluorescence (SIF) has shown great potential to monitor the photosynthetic activity of terrestrial ecosystems. However, several issues, including low spatial and temporal resolution of the gridded datasets and high uncertainty of the individual retrievals, limit the applications of SIF. In addition, inconsistency in measurements footprints also hinder the direct comparison between gross primary production (GPP) from eddy covariance (EC) flux towers and satellite-retrieved SIF. In this study, by training a neural network (NN) with surface reflectance from the MODerate-resolution Imaging Spectroradiometer (MODIS) and SIF from Orbiting Carbon Observatory-2 (OCO-2), we generated two global spatially contiguous SIF (CSIF) datasets at moderate spatio-temporal resolutions (0.05 degree 4-day) during the MODIS era, one for clear-sky conditions (2000-2017) and the other one in all-sky conditions (2000-2016). The clear-sky instantaneous CSIF ($CSIF_{clear-inst}$) shows high accuracy against the clear-sky OCO-2 SIF and little bias across biome types. The all-sky daily average CSIF ($CSIF_{all-daily}$) dataset exhibits strong spatial, seasonal and interannual dynamics that are consistent with daily SIF from OCO-2 and the Global Ozone Monitoring Experiment-2 (GOME-2). An increasing trend (0.39%) of annual average $CSIF_{all-daily}$ is also found, confirming the greening of Earth in most regions. Since the difference between satellite observed SIF and CSIF is mostly caused by the environmental down-regulation on $SIF_{yield}$, the ratio between OCO-2 SIF and $CSIF_{clear-inst}$ can be an effective indicator of drought stress that is more sensitive than normalized difference vegetation index and enhanced vegetation index. By comparing $CSIF_{all-daily}$ with gross primary production (GPP) estimates from 40 EC flux towers across the globe, we find a large cross-site variation (c.v. = 0.36) of GPP-SIF relationship with the highest regression slopes for evergreen needleleaf forest. However, the cross-biome variation is relatively limited (c.v. = 0.15). These two contiguous SIF datasets and the derived GPP-SIF relationship enable a better understanding of the spatial and temporal variations of the GPP across biomes and climate.

## 1 Introduction

Obtaining a spatio-temporal continuous photosynthetic carbon fixation or gross primary production (GPP) dataset is crucial to food security, ecosystem service and health evaluation, and global carbon cycle studies (Beer et al., 2010). However, this is

not possible without remote sensing data, since *in situ* carbon flux measurements, such as FLUXNET (Baldocchi et al., 2001), are usually costly and have limited spatial and temporal coverages (Schimel et al., 2015). Many remote sensing based productivity efficiency models (PEMs) have been built, but the model structure and parameterizations differ from each other and the performance of most models is not satisfactory in terms of simulated inter-annual variability and trends (Anav et al., 2015; Chen et al., 2017).

Müller (1874) found that the chlorophyll fluorescence (ChlF) from a dilute chlorophyll solution was much stronger than the ChlF from a green leaf, suggesting that an alternative energy pathway exists for leaves *in vivo*. In the 1980s, scientists found that plant photosynthesis and heat dissipation are two alternatives to quench the excited chlorophylls, and there is a close linkage between ChlF and carbon assimilation rate (Genty et al., 1989; Krause and Weis, 1991). Leaf-level photosynthesis ($A_{leaf}$) and fluorescence (ChlF) share the same source of energy originating from photosynthetically active radiation (PAR) absorbed by chlorophyll ($APAR_{chl}$), which can be written using a light use efficiency approach (Monteith, 1972):

$$ChlF = PAR \times fPAR_{chl} \times \phi_F \quad (1)$$

$$A_{leaf} = PAR \times fPAR_{chl} \times \phi_P \quad (2)$$

where $\phi_F$ and $\phi_P$ represent the efficiencies for ChlF emission and photochemistry, respectively. $fPAR_{chl}$, being different from the conventional definition of fraction of photosynthetically active radiation absorption, only considers the fractions absorbed by chlorophyll pigments where the photosynthesis and fluorescence originate (Zhang et al., 2018c). However, ChlF measurements have been mostly conducted at the leaf level, using pulse amplitude modulation (PAM) fluorometers (Porcar-Castell et al., 2008; Roháček and Barták, 1999). In this case, the measured ChlF intensity is not induced by the sun but by the modulated light source. Although the absolute value of the ChlF intensity does not directly link to $A_{leaf}$, it can still be used to calculate the fluorescence yield and investigate the reaction mechanism of the energy partitioning during the light reaction, and to calculate the quantum yield for photochemistry or as tool to detect plant reactions under stress (Adams and Demmig-Adams, 2004; Flexas et al., 2002).

The successful retrieval of solar-induced (steady-state) chlorophyll fluorescence (SIF) from satellites have made it possible for vegetation photosynthetic activities to be observed at the global scale (Frankenberg et al., 2011; Guanter et al., 2012; Joiner et al., 2011, 2013). Satellite SIF can be expressed as a function similar to the ChlF at the leaf level but with extra terms considering the radiative transfer within the canopy and through the atmosphere (Joiner et al., 2014):

$$SIF_{sat}(\lambda) = PAR \times fPAR_{chl} \times \Theta_F(\lambda) \times f_{esc}(\lambda, \theta_s, \theta_v, \phi) \times \tau_{atm}(\lambda, \theta_s, \theta_v, \phi) \quad (3)$$

where the satellite retrieved SIF ($SIF_{sat}$), fluorescence yield ($\Theta_F$), $f_{esc}$, $\tau_{atm}$ are all functions of the wavelength ($\lambda$), in addition, $f_{esc}$ and $\tau_{atm}$ are also affected by sun-sensor geometry characterized by sun zenith angle ($\theta_s$), view zenith angle ($\theta_v$), relative azimuth angle ($\phi$). $f_{esc}$ is a factor describing how much SIF emitted by the chloroplast leaves the canopy, and $\tau_{atm}$ is a function of atmospheric optical depth, which indicates how much SIF that leaves the canopy top passes through the atmosphere before it is captured by the satellite sensors. It should be noted that the fraction of PAR for fluorescence ($fPAR_F$) may have

different activation spectrum than that for photosynthesis ($fPAR_{chl}$), but this difference is ignored here for simplicity. Although additional factors come into play during this process, satellite retrieved SIF shows high consistency with GPP using both model simulations and ground-based measurements from eddy covariance (EC) flux towers, at least at the monthly time scale (Guanter et al., 2014; Li et al., 2018a; Zhang et al., 2016c, 2016b). In addition, recent studies suggest that the GPP-SIF relationship is consistent across biome types (Sun et al., 2017). This finding, if valid across all biomes, would greatly benefit the usage of SIF for model benchmarking (Luo et al., 2012) and global GPP estimation.

However, several issues hinder exploring the relationship between SIF and *in situ* GPP estimates. Since the SIF signal is very small and sensors used to retrieve SIF were not initially built to estimate SIF, the satellite-retrieved SIF usually has a large footprint and large uncertainties in individual retrievals (Frankenberg et al., 2014; Joiner et al., 2013, 2016). For instance, the SIF retrieval from Global Ozone Monitoring Experiment-2 (GOME-2) has a footprint of 40 km×40 km or larger; and the SIF from Greenhouse gases Observing SATellite (GOSAT) has a circular footprint with 10.5 km in diameter. Direct comparison between the satellite retrieved SIF signal and GPP estimates from EC flux tower sites thus faces the problem of spatial inconsistency except in areas of large homogenous landscape, e.g., the US Midwest cropland (Zhang et al., 2014) or boreal evergreen forests (Walther et al., 2016). However, corn (C4 pathway) and soybean (C3 pathway) in SIF footprints have different electron use efficiencies (Guan et al., 2016), which should affect the relationship between SIF and GPP. The low precision of SIF measurements also leads to a need for averaging multiple pixels either in space or time before being used.

SIF retrieved from the Orbiting Carbon Observatory-2 (OCO-2) satellite partially solved this issue with a much smaller footprint size (1.3 km×2.25 km), higher signal to noise ratio compared to GOSAT (relatively higher SIF retrieval accuracy) and much larger numbers of observations per day (Frankenberg et al., 2014; Sun et al., 2018). However, due to the sparse sampling strategy and long revisit cycle, the OCO-2 SIF data have large gaps between nearby swaths and the average sampling frequency for each flux tower site is only 3.21/year during 2015-2016 (Lu et al., 2018). In addition, OCO-2 is often aggregated to monthly dataset at relatively coarse spatial resolution, typically at 1° × 1°, which limits its application in small regions. Although several statistical methods have been proposed to downscale satellite observations to finer spatial-temporal resolutions (Tadić et al., 2015, 2017), considering the large land surface heterogeneity and wide gaps between OCO-2 swaths (~ 100 km), it could be challenging to apply these methods to OCO-2 SIF.

A high spatio-temporal resolution SIF dataset is needed to improve our understanding of the relationship between SIF and GPP and provide accurate GPP estimates at the global scale. As discussed previously, the satellite-observed SIF contains signals from $APAR_{chl}$, fluorescence yield, and canopy and atmospheric attenuation. $APAR_{chl}$ is considered to be the first order approximation of SIF as it exhibits high correlation with SIF at the canopy scale (Du et al., 2017; Rossini et al., 2016; Verrelst et al., 2015; Zhang et al., 2018c). Previous studies have shown that $fPAR_{chl}$ can be inversely estimated using the surface reflectances and radiative transfer models (Zhang et al., 2005, 2016a). The canopy structure information that affects the SIF

reabsorption within canopy is also embedded in the near infrared reflectance (Badgley et al., 2017; Knyazikhin et al., 2013; Yang and van der Tol, 2018). Many previous studies have shown high correlation between SIF and vegetation indices (VIs), especially VIs related to the chlorophyll concentration (Frankenberg et al., 2011; Guanter et al., 2012). Therefore, broad-band surface reflectances may have the potential to be used to estimate vegetation information and reconstruct global SIF (Duveiller

and Cescatti, 2016; Gentine and Alemohammad, 2018a). However, physical models that can predict SIF (e.g. the Soil Canopy Observation, Photochemistry and Energy fluxes, SCOPE (van der Tol et al., 2009)) often require many parameters, making it difficult to use reflectance and modelling to predict SIF at a larger scale.

Neural networks (NN), together with many other machine learning algorithms, have been used with remote sensing datasets

in the Earth sciences, especially for carbon and water fluxes estimation (Alemohammad et al., 2017; Jung et al., 2011; Tramontana et al., 2016), land cover mapping (Kussul et al., 2017; Zhu et al., 2017), soil moisture retrievals and downscaling (Alemohammad et al., 2018; Kolassa et al., 2018) or to bypass parameterization (Gentine et al., 2018). These studies mostly attempted to link the satellite signals with limited *in situ* observation or model simulations for model training, while taking advantage of the large amount of data in remote sensing observations; they applied the trained algorithm to generate a regional

or global dataset. Reconstructing SIF from surface reflectance, on the other hand, uses no *in situ* observations but faces more problems related to the satellite data quality assurance. The SIF-reflectance relationship is complicated, and the NN benefits from the fact that an explicit physical and radiative transfer relationship is not required.

In this study, we aim to generate a global contiguous SIF (CSIF) product based on the SIF retrievals from OCO-2 and surface

reflectances from Moderate-resolution imaging spectroradiometer (MODIS) onboard Terra and Aqua satellite. The CSIF dataset aims to fill the spatial gaps between the OCO-2 swaths and temporal gaps due to the long revisit cycle of OCO-2. Specifically, we first trained and validated the NN using the satellite observed instantaneous SIF under clear-sky conditions so that the relationship is not affected by cloud-related artifacts. We further generated two SIF products, namely the clear-sky instantaneous SIF ($CSIF_{clear-inst}$) and the all-sky daily SIF ($CSIF_{all-daily}$). The spatio-temporal variations of these CSIF products

were analyzed and compared with SIF from OCO-2 and three other GOME-2 SIF datasets. Finally, we showed two applications of CSIF datasets: (1) monitoring drought impact using $CSIF_{clear-inst}$ and OCO-2 SIF; (2) evaluating the GPP-SIF relationship by comparing CSIF with GPP estimates from 40 flux tower sites.

## 2 Materials and methods

### 2.1 OCO-2 solar-induced chlorophyll fluorescence dataset

The 8100r OCO-2 SIF data between September 2014 to December 2017 were used for NN training and evaluation (Frankenberg, 2015; Frankenberg et al., 2014; Sun et al., 2018). The daily sounding-based SIF retrievals at 757 nm were first aggregated to 0.05-degree (around 5.6 km×5.6 km at equator), consistent with MODIS climate model grid (CMG) resolution.

The reasons for using this resolution include: (1) it is directly comparable (in the same order of magnitude) to the OCO-2 SIF footprint size (around 1.3km×2.25km) and the samples within each gridcell can be more evenly distributed and, thus, more representative of the gridcell SIF values than using much coarser $1° \times 1°$ or $2° \times 2°$ grids; (2) by averaging multiple observations, the uncertainty in the SIF signal can be approximately reduced by a factor of $\sqrt{n}$ ($n$ is the number of observations

within this gridcell), assuming independent estimates and homogeneous SIF value within each gridcell (Frankenberg et al., 2014). During this aggregation, we only used cloud-free observations indicated by the OCO-2 cloud flag. For each 0.05-degree gridcell, the SIF value was only calculated when it contained more than 5 cloud-free SIF soundings. Although several studies have shown that SIF at different wavelengths has different sensitivity to stress and leaf and canopy reabsorption (Porcar-Castell et al., 2014; Rossini et al., 2015, 2016), we only use SIF at 757nm since it showed superior performance than SIF at 771nm in

predicting GPP (Li et al., 2018a). The years 2015 and 2016 were used for training and 2014 and 2017 were used for validation. Altogether 2947819 SIF gridcells passed quality check during 2014-2017. Figure 1 shows the spatial distribution of the SIF gridcells used for training and validation (test). It should be noted that the OCO-2 satellite starts to obtaining data from September 2014 and experienced some malfunctioning during August and September in 2017, causing lower coverage for validation samples in boreal regions.

In addition to these cloud-free observations, we also calculated the all-sky SIF at 0.05-degree resolution. All SIF retrievals that passed the suggested quality checks (documented in detailed by Sun et al. (2018)) were used for the aggregation. The aggregated all-sky instantaneous SIF retrievals were converted to daily values based on the solar zenith angle (Zhang et al., 2018a). We used this dataset to validate the all-sky daily SIF ($CSIF_{all-daily}$) (see section 2.5). In both cloud free and all-sky

aggregations, only observations from the nadir mode were used since glint mode tends to underestimate SIF (Sun et al., 2018).

**2.2 MODIS reflectance dataset (MCD43C4 V006)**

We used the 0.05-degree daily Nadir Bidirectional reflectance distribution Adjust Reflectance (NBAR) product from MODIS (MCD43C4 V006) during 2000-2017 as input variables for the NN. The NBAR product compute the reflectance at a nadir viewing angle for each pixel at local solar noon. Compared to MOD09 or MYD09 surface reflectance product, it removed the

angle effects, and therefore, should be more stable and consistent (Schaaf et al., 2002). This dataset was processed in two different ways for training and prediction. For the training process, following (Gentine and Alemohammad, 2018a), we extracted the reflectance from the first four bands of MODIS (centered at 645nm, 858nm, 469nm and 555nm, respectively) for the corresponding pixels and days when the cloud-free SIF observations were obtained. It should be noted that although the MCD43C4 is generated for each day and can match the daily SIF observations, the MCD43C4 NBAR uses 16-day worth

of inputs and so that the reflectances includes the information of other days than the day of interest. However, we consider this to have limited effects since: (1) the vegetation growth/changes are continuous in time, (2) the NBAR product uses 16-day of data but also emphasizes the specific day of interest (Schaaf, 2018). These four bands were selected because the visible and near-infrared band included most of the vegetation information and drives the variation of SIF (Verrelst et al., 2015). We also

tested using all 7 band with/without the meteorological variables (temperature and vapor pressure deficit, obtained from the OCO-2 SIF lite files) to train the NN, but the improvement in training and validation were very minor ($R^2$ increased by less than 0.01, data not shown) and thus we decided not to use it. Since SIF is very sensitive to the incoming solar radiation, using cloud-free training samples can minimize the uncertainty of using cosine of the solar zenith angle as the proxy of incoming PAR. It should be noted that the training dataset may contain snow-affected samples, but these were not removed to get a more realistic prediction of SIF during winter.

For prediction, we first aggregated the daily reflectance to 4 days. The 4-day temporal resolution is selected to reach a balance among application requirements, information redundancy and dataset sizes. During this process, we used a gap-filling and smoothing algorithm to reconstruct the surface reflectance for the four bands. The detailed description of the gap-filling algorithm can be found in Zhang et al. (2017a). In this study, we slightly modified the algorithm by not applying the Best Index Slope Extraction (BISE) algorithm and Savitzky-Golay (SG) filter. The reconstructed 4-day 0.05-degree reflectance together with other datasets allowed us to predict SIF at 4-day 0.05-degree resolution during 2000-2017. Since this processing does not involve any extra information and only uses the reflectance observations from the successful model inversion, it should be comparable to the reflectance used for NN training.

### 2.3 Machine learning algorithms

A feedforward neural network (NN) is a number of computational nodes (called neurons) structured in a single or multi-layer architecture. Each neuron is connected with all neurons in the previous layer and next layer. The neuron values are calculated using an activation function with a pre-activated value, i.e., the weighted sum of all neurons in previous layer plus biases. The training of the NN attempts to optimize these weights and biases so that the differences between the output variable in the training data and NN prediction is minimized. In this study, we used Tensorflow (https://www.tensorflow.org) and built feedforward networks with 1-3 layers and 2-9 neurons for each layer. After training models with data from 2015 and 2016, we validated the models using the test dataset from year 2014 and 2017. We then picked the one with best performance and simplest structure for SIF prediction. The rectified linear unit (ReLU) was used as the activation function since it has shown better performance in our application and the cost function used is the root-mean-square error (RMSE). We used 50 epochs with a batch size of 1024. Before training, each variable was normalized by its mean and standardized deviation. Since the NN is not deep and there is no sign of overfitting, we did not use any regularization methods during the training.

### 2.4 Reconstructing the clear-sky instantaneous SIF and daily SIF

During the NN training process, we only used the SIF and reflectance data in clear-sky conditions, and therefore cos(SZA) was used as a proxy of the incoming photosynthetically active radiation at top-of-canopy. In the prediction process, we also used the calculated cos(SZA) based on the satellite overpass local solar time and latitude. Since we did not consider the cloud and aerosol attenuation of the PAR, this product was referred to as the "clear-sky instantaneous SIF ($CSIF_{clear-inst}$)".

In addition to the clear-sky instantaneous SIF, we also calculated two daily SIF data by assuming that the incoming solar radiation is the only factor that drives the diurnal cycle (Zhang et al., 2018a). All-sky daily SIF (CSIF$_{all-daily}$) can be calculated using the clear-sky top-of-canopy radiation (PAR$_{clear-inst}$) and the daily average radiation from Breathing Earth System
Simulator (BESS) (Ryu et al., 2018):

$$CSIF_{all-daily} = \frac{CSIF_{clear-inst}}{PAR_{clear-inst}} \times PAR_{daily}^{BESS} \qquad (4)$$

where PAR$_{clear-inst}$ was calculated following previous studies that only considered atmospheric scattering (see Appendix A1). Clear-sky daily SIF (CSIF$_{clear-daily}$) assumes no cloud throughout the day and can be calculated by multiplying CSIF$_{clear-inst}$ with a daily correction factor ($\gamma$) (Zhang et al., 2018a):

$$CSIF_{clear-daily} = CSIF_{clear-inst} \times \gamma \qquad (5)$$

$\gamma$ is calculated as the ratio between the cos(SZA) during the satellite overpass and the daily averaged cos(SZA).

## 2.5 GOME-2 SIF (SIF$_{GOME-2}$), Reconstructed SIF from GOME-2 (RSIF$_{GOME-2}$) and SIF* datasets.

In this study, we also used the GOME-2 SIF (SIF$_{GOME-2}$), reconstructed SIF from GOME-2 (RSIF$_{GOME-2}$) using machine learning and SIF* dataset in comparison with our contiguous SIF from OCO-2. The GOME-2 SIF V27 was retrieved using a
principle component analysis algorithm in the wavelength range 734-758 nm (Joiner et al., 2013, 2016). The V27 version, compared to the widely used V26, provides daily correction factor and improved bias correction and calibration (https://avdc.gsfc.nasa.gov/pub/data/satellite/MetOp/GOME_F/). The level 3 monthly 0.5-degree daily-average-SIF was used to compare with CSIF$_{all-daily}$.

RSIF$_{GOME-2}$ (Gentine and Alemohammad, 2018a) uses a similar machine learning technique approach to CSIF but the training is based on the bi-weekly gridded SIF product from GOME-2, and 8-day MYD09A1 reflectance dataset. Both clear-sky and cloudy-sky SIF are used for NN training. This dataset has a spatial resolution of 0.05-degree and 8-day temporal resolution. Both RSIF$_{GOME-2}$ and CSIF$_{all-daily}$ were aggregated to the 0.5-degree and semi-month to facilitate the comparison.

SIF* dataset (Duveiller and Cescatti, 2016) applies a statistical method and calibrates a model that links monthly 0.5-degree SIF to NDVI, evapotranspiration (ET) and land surface temperature (LST) dataset for each moving window. The model and its spatio-temporally varied parameters were then applied to finer resolution dataset (NDVI, ET, LST) with a weighted average to generate SIF at 0.05-degree resolution. In this study, we used the 0.5-degree monthly SIF* dataset during 2007-2013 to compare with CSIF.

## 2.6 Comparing CSIF with GPP at flux tower sites

We further compared the CSIF dataset to GPP estimates from the Tier 1 FLUXNET2015 datasets (http://fluxnet.fluxdata.org) to investigate the SIF-GPP relationship. Since the CSIF dataset is continuous in space and time, it provides many more samples pairs compared to the original OCO-2 SIF data (Lu et al., 2018). However, because of the landscape heterogeneity and inconsistency between the flux tower footprint and CSIF pixel size, a rigorous site selection is needed. We took the vegetation growth condition into consideration during this process: (1) the annual average, minimum, maximum and seasonal variability (represented by standard deviation) of normalized difference vegetation index (NDVI, from MOD13Q1 C6) for the target pixel (where the flux tower locates, 250 m by 250 m) need to be similar (within 20% difference or 0.05 NDVI) with the neighboring (5 km by 5 km) area; (2) Maximum NDVI value for target pixel and neighboring area need to greater than 0.2 (not barren). The daily GPP estimates, estimated using nighttime method (Reichstein et al., 2005) were averaged and aggregated into 4-day values to compare with CSIF. 4-day GPP based on more than 80% of half-hourly valid (not gap-filled) net ecosystem exchange was retained. Only sites that have at least 92 valid observations (1 year) were used. Only 40 out of 166 sites passed these criterions and were grouped into different biome types (Table S1). In addition to $CSIF_{all\text{-}daily}$, we also calculated $CSIF_{clear\text{-}daily}$ and $CSIF_{site}$ which used flux tower observed radiation instead of $PAR_{daily}^{BESS}$ in Eq. (4).

## 3 Results

### 3.1 NN training and validation

The NN with one layer and five neurons generally predicts the OCO-2 SIF during the training with a coefficient of determination ($R^2$) around 0.8, and an RMSE of 0.18 mW m$^{-2}$ nm$^{-1}$ sr$^{-1}$ (Figure 2). The model also performs well in the validation ($R^2$=0.79, RMSE=0.18) and does not show effects of overfitting. Using a variety of layer (1-3) and neurons (2-9) combinations, we found that 1-layer with 5-neurons exhibited slightly higher model performance during the validation compared with more complex NN (Figure A1). Therefore, we chose to use the 4-band reflectances to feed the one-layer-five-neuron NN to generate the contiguous SIF for 2000 to 2017 when MCD43C4 NBAR dataset is available.

We also investigated the bias of our prediction among different biome types in Figure 3. For 9 out of 14 biome types, the differences between the $CSIF_{clear\text{-}inst}$ and the satellite-retrieved SIF are less than 10%; and most of the biases were within 5%. Wetlands and urban ecosystem show a 15% bias compared to the satellite retrieved SIF, which may be caused by the water or built-up contamination on the reflectance signal and the relatively small sample numbers. For savannas and grassland, the changes in fluorescence yield due to seasonal drought may be important, which cannot be considered in the NN based on reflectances only. Over croplands, CSIF exhibits a 12% underestimation. The croplands usually have high nitrogen/chlorophyll concentration that may not be fully captured by the four broad-band reflectances (Wu et al., 2008). Because we did not build biome-specific NNs for the training, we do not expect biome-specific (especially needle leaf vs. broad leaf) relationships between SIF and reflectance. Interestingly, we still reproduced SIF with very high accuracy regardless of the plant function

traits (PFT), i.e., leaf types and canopy characteristics (leaf clumping, etc.). This suggests that the escape factor and long-term changes in mean fluorescence yield might be correctly accounted for by the NN across PFTs, through the information available in the reflectances only. However, it should be noted that this does not suggest that the NN and reflectances can fully replicate the fluorescence yield variations due to short-term variations caused by stresses.

We also compared the times series of predicted CSIF and OCO-2 SIF for 12 typical biome types (Figure 4). The predicted CSIF accurately captures the seasonal and interannual variation for most biome types, while the standard deviation for each DOY is usually smaller than OCO-2 SIF. This may suggest that the uncertainty of SIF is smaller in CSIF dataset. For some ecosystems, e.g., DBF, MF, and CRO, CSIF shows slight underestimation during the peak growing season.

When comparing the daily average SIF from satellite retrievals with the predicted all-sky daily CSIF ($CSIF_{all-daily}$) dataset (Figure 5), the predicted SIF exhibits ~7% underestimation, with an $R^2$ of 0.71 and a RMSE of 0.08 mW m$^{-2}$ nm$^{-1}$ sr$^{-1}$. The clear-sky daily CSIF ($CSIF_{clear-daily}$) shows ~11% overestimation, with a slightly higher $R^2$ and lower RSME. Considering the uncertainty in SIF retrievals and the inconsistency in time of the comparison (satellite SIF was based on instantaneous PAR at

the time of satellite overpass and converted to daily values assuming the atmospheric condition did not change within a day, predicted CSIF was based on 4-day average PAR), the all-sky daily CSIF performs reasonably well.

### 3.2 Spatial temporal variation of the global 0.05-degree SIF datasets

Using the trained NN with the gap-filled reflectance datasets, we produced two global CSIF datasets at 4-day temporal and 0.05-degree spatial resolution. Figure 6 shows the spatial patterns of the 90 percentile for each pixel and the annual average

for both clear-sky instantaneous CSIF ($CSIF_{clear-inst}$) and the all-sky daily average CSIF ($CSIF_{all-daily}$). For the 90 percentile, $CSIF_{clear-inst}$ exhibits hotspots in the tropical rainforest, south Asia, and North America Corn belt, consistent with regions with high peak productivity (Guanter et al., 2014); $CSIF_{all-daily}$ shows similar spatial patterns, but with relatively lower values in the tropical forest, due to the persistent cloud coverage. For the annual average SIF, tropical forests exceed temperate cropland and show very high values for instantaneous clear-sky SIF. In all conditions, African tropical forests exhibit lower values than

Amazon and Southeast Asia tropical forests.

We further investigated the seasonal and interannual variation of the all-sky daily SIF across the latitudes. The tropical regions show continuous high SIF values across seasons and the northern mid- to high-latitude regions also exhibit recurrent high values during the northern hemisphere summers (Figure 7a). Near 40°S a hot spot is present in austral Summer, with high

interannual variability. Low SIF values can be found in dry years (2006-2007, 2009-2010) while high values were observed in wet or normal years (2010-2011, 2012-2015). The global average SIF also displays a strong seasonality coinciding with the North Hemisphere growing season (Figure 7b). For the annual total SIF values, a statistically significant increasing trend

(Mann-Kendall test, $p<$1e-4) is found with around 0.39% increase per year. Year 2015 exhibited a low anomaly after detrending, which may be caused by the El Niño events (Figure 7c).

The spatial pattern of the trend in CSIF$_{all-daily}$ is displayed in Figure 8. Increasing trend dominates Europe, southeast Asia and
south Amazon. Decreasing trend is mostly found in east Brazil, east Africa and some area inland Eurasia. The histogram also shows a positive shift with a magnitude (0.00027 mW nm$^{-1}$ sr$^{-1}$ yr$^{-1}$) similar to the average global trend in Figure 7c. The spatial pattern of CSIF$_{all-daily}$ is very similar with the trend pattern of MODIS EVI (C6) (Zhang et al., 2017b), but the south Brazilian Amazon forest shows a more positive trend than that of EVI.

### 3.3 Comparison between SIF from GOME-2 and CSIF

We then compared the CSIF datasets with the reconstructed SIF (RSIF) and SIF* based on coarser-scale and all-sky GOME-2. Although these datasets were trained based on different satellites, the relationship between CSIF and RSIF or CSIF and SIF* is consistent across most regions across the globe (Figure 9). The $R^2$ values are generally high (> 0.8) for most regions except over tropical rainforests, barren regions in western US, northwestern China and northern Canada and Russia. The low $R^2$ values are mostly due to the relatively low variability in the temporal domain in the tropics but are also indicative of regions
strongly polluted by cloud cover in which CSIF might have a competitive advantage, as the training OCO-2 data better observes the surface due to smaller footprint and with higher signal to noise ratio. The regression slopes are higher for regions with persistent cloud cover (e.g., tropical forest). In the time series comparison (Figure 9e-p), all three SIF datasets show similar seasonal patterns, while GOME-2 based RSIF and SIF* generally show higher values than CSIF. In addition, RSIF exhibits larger fluctuation during the non-growing season for some sites, which may be caused by snow contamination.

We further compared the CSIF$_{all-daily}$ with GOME-2 daily average SIF (Figure 10). In general, the correlation is much lower as compared with RSIF for most regions. For regions with high variability in temporal domain, the CSIF$_{all-daily}$ still shows high $R^2$ values with respect to GOME-2 SIF. The regression slopes exhibit smaller variation except for the Amazonian tropical rainforests, southeast Asia, and barren regions in Sahara, western US, northwestern China, central Australia and Andes
mountains in South America. In general, considering the various uncertainties and different satellite overpass times, sensors used, and retrieval algorithms, CSIF$_{all-daily}$ well captured the GOME-2 SIF variations both in space and time. In addition, since GOME-2 SIF in most Argentina is affected by the South American Anomaly (SAA), the coefficient of determination values are also lower as compared with Figure 9.

### 3.4 Using CSIF for drought monitoring

Since the CSIF dataset only uses broadband reflectances, it should not contain the SIF$_{yield}$ information. Compared to the SIF retrieved from OCO-2, the difference can be mostly attributed to the SIF$_{yield}$. Therefore, the difference or ratio between SIF$_{OCO-}$

$_2$ and CSIF can reflect the environmental stress on $SIF_{yield}$. Figure 11 shows the difference between instantaneous clear-day OCO-2 SIF and $CSIF_{clear-inst}$. Except for Figure 11c, the difference mostly captures the physiological limitation of drought on energy partitioning after being absorbed by chlorophyll. The spatial extent of drought is also well-captured by the difference, where the most severe drought impacted places also exhibited the largest decline (e.g., Namibia, Botswana, Zimbabwe in (a),

northeast Amazon in (b) and southern Spain, south most France, central Italy, Croatia and Bosnia and Herzegovina). The drought impact on California is less pronounced, possibly because of the irrigation systems and sparse sampling points.

We further focused on the 2015 European drought to compare the drought response of CSIF and two vegetation indices (normalized difference vegetation index, NDVI; enhanced vegetation index, EVI). Because the OCO-2 samples were not

collected at the same swath for each DOY, a large fluctuation can be found in OCO-2 SIF and on the CSIF (which are using the same pixels for a fair comparison) (Figure 12a-d). However, when calculating the ratio between CSIF and OCO-2 SIF, its variation can be mostly attributed to the variation in $SIF_{yield}$, which can quantify the drought stress on plant physiology. In all three regions, the ratio between OCO-2 SIF and CSIF experienced a decrease during the drought period, but the signal is only obvious after applying a smoothing filter. The two vegetation indices, NDVI and EVI, on the other hand, show a reduced

response in Spain and Italy, perhaps due to the plants adaption or very short drought duration.

### 3.5 GPP-CSIF relationship across biome types

With this contiguous $SIF_{all-daily}$ dataset, we finally evaluated the GPP-CSIF relationship using GPP estimates from 40 flux tower sites from FLUXNET tier 1 dataset. The regression slope between GPP and CSIF ($a_{GPP/CSIF}$) spreads across sites with a regression slope ranging from 11.91 to 68.59 (g C m$^{-2}$ day$^{-1}$/mW m$^{-2}$ nm$^{-1}$ sr$^{-1}$) for $CSIF_{all-daily}$, 11.61 to 72.10 (g C m$^{-2}$ day$^{-1}$

$^{-1}$/mW m$^{-2}$ nm$^{-1}$ sr$^{-1}$) for $CSIF_{site}$ and 11.37 to 62.75 (g C m$^{-2}$ day$^{-1}$/mW m$^{-2}$ nm$^{-1}$ sr$^{-1}$) for $CSIF_{clear-daily}$. The R$^2$ value for each individual site ranges from 0.01 to 0.93 with a median value of 0.64, 0.62 and 0.69 for all-daily, site, and clear-daily CSIF, respectively. The RMSE is 1.67 g C m$^{-2}$ day$^{-1}$ on average.

Although the CSIF-GPP relationship varies across 40 sites, when lumping all observations within each biome type, the

variation is smaller (c.v. = 0.16, rhombus in Figure 13c,f,i). Specifically, ENF exhibited a significant larger $a_{GPP/CSIF}$ (two-tiled student's t test, p=0.036), which is caused by a stronger canopy reabsorption/scattering of SIF. OSH only have one site and also showed very high value. If both biomes are eliminated, the $a_{GPP/CSIF}$ for rest biomes exhibited smaller variation (c.v. = 0.08).

The CSIF-GPP relationship not only varies across biome, but also varies within each biome type, especially for evergreen needleleaf forest (ENF, 9 sites), grassland (GRA, 8 sites) and wetland (WET, 2 sites) (Figure 13c,f). For $CSIF_{all-daily}$, the average within-biome variation of $a_{GPP/CSIF}$ (c.v. = 0.26±0.08) is comparable to cross-sites variations (c.v. = 0.34), but larger than the

cross-biome variations (c.v. = 0.16, using the biome-specific CSIF-GPP factor). Similar pattern can be found using CSIF$_{site}$ or CSIF$_{clear-daily}$.

## 4 Discussion

### 4.1 Information in contiguous SIF produced by machine learning

Vegetation photosynthetic activity has variations in several respects controlled by vegetation type, phenology, coverage, and interactions with the environment. These variations can be expressed in the spatial, seasonal, diurnal and/or interannual domains (Zhang et al., 2018a). Machine learning algorithms, try to minimize the differences between the predicted SIF and the satellite observed SIF. For OCO-2 SIF and the MODIS reflectance used for NN training, the variance in the spatial and seasonal domains are largest. Therefore, the NN generally predicts SIF well in these two domains. The interannual variations (i.e., the variations caused by year to year anomalies, e.g. due to drought) typically have much smaller variance and is more difficult to capture. This is why some machine learning products fail to reproduce interannual variability accurately (Jung et al., 2011). Using additional variables that is sensitive to this interannual anomaly in the model training can improve the model performance (Alemohammad et al., 2017; Gentine and Alemohammad, 2018b; Tramontana et al., 2016).

In this study, since the variations in SIF$_{yield}$ are relatively small (Lee et al., 2015), and cannot be detected by broadband surface reflectances, the SIF$_{yield}$ information may not be reproduced by our CSIF data. Because the environmental limitation on SIF$_{yield}$ may be complicated (may not be a linear combination of temperature, VPD or surface reflectance in the shortwave infrared) and biome specific (van der Tol et al., 2014), inclusion of other environmental variables and reflectances in shortwave bands during NN training did not greatly increase the SIF prediction accuracy. It should also be noted that SIF$_{yield}$ is relatively stable when no strong environmental limitation is present (Zhang et al., 2018c). Therefore, the CSIF product should be considered as a good proxy of OCO-2 SIF.

The satellite-retrieved SIF has a relatively large uncertainty for each individual sounding, typically ranging between 0.3-0.5 mW m$^{-2}$ nm$^{-1}$ sr$^{-1}$ (Frankenberg et al., 2014). Previous site-level studies usually use SIF averaged over a large buffered area (Li et al., 2018a; Verma et al., 2017) to reduce the uncertainty. Assuming the uncertainty is unbiased and has a Gaussian distribution, machine learning algorithms are designed to reproduce SIF with lower uncertainty. Compared with previous studies that use light use efficiency models to downscale SIF to higher resolution (Duveiller and Cescatti, 2016), this study does not rely on multiple modeled input (evapotranspiration for example) that may introduce additional uncertainties.

We also found a significant increasing trend (0.39% year$^{-1}$) in the global annual CSIF$_{all-daily}$ (Figure 7). This trend is close to the GPP trend derived from the satellite-data driven vegetation photosynthesis model (VPM) (0.32% year$^{-1}$) (Zhang et al.,

2017a), but much greater than GPP derived from other remote sensing data-driven models (FluxCOM 0.01% year$^{-1}$ (Tramontana et al., 2016), BESS GPP 0.22% year$^{-1}$ (Jiang and Ryu, 2016), MODIS C6 0.26% year$^{-1}$ (Zhao et al., 2005), and WECANN -0.8% year$^{-1}$ [affected by the decreasing GOME-2 SIF trend (Zhang et al., 2018b)] (Alemohammad et al., 2017)). Considering there is no significant trend (-0.02% year$^{-1}$, $p>0.1$) in BESS PAR (Ryu et al., 2018), this increase is likely caused by the greening of the Earth (Zhang et al., 2017b; Zhu et al., 2016) as captured in the MODIS reflectance data. This increasing trend is also within the range of most Earth system models' predictions (Anav et al., 2015). We also observed a more pronounced increasing trend in southern Amazon than using MODIS EVI (Zhang et al., 2017b). This may suggest that CSIF is less likely to suffer from high biomass saturation than optical vegetation indices and can more effectively detect changes in tropical rainforests or over high leaf area regions such as croplands.

## 4.2 The use of satellite SIF for drought monitoring

Drought can be categorized into different stages. At an early stage, when plants sense water deficit in the soil and higher vapor pressure deficit in the atmosphere, they reduce water loss through stomatal closure. This, in turn, also reduces the $CO_2$ exchange from stomatal closure and inhibits photosynthesis. The quantum yield for heat dissipation will increase accompanied with a decrease in quantum yield for photochemical quenching and fluorescence (Genty et al., 1989; Porcar-Castell et al., 2014). This should allow satellites to potentially capture this decrease in the SIF signal (especially during the mid-noon when stress is more pronounced) as an indicator of vegetation stress. In the second stage, with prolonged dry conditions, plants will recycle the nitrogen in the leaves as represented by a decrease of the greenness (chlorophyll content) of leaves. In the third stage, if the drought continues, leaf senescence and vegetation mortality may follow. SIF can potentially detect changes during all those drought stages, whereas broadband reflectances based indices (NDVI, EVI) should only see the second and third stages.

Previous drought monitoring studies have mostly used vegetation indices (VIs) as a indictor of drought stress (Ji and Peters, 2003; Zhang et al., 2013). However, vegetation indices can only respond to drought changes in the plants' optical properties (mostly during the second and third stages). For most plants, there might be a tipping point where plants will not recover from drought-induced xylem cavitation (Urli et al., 2013). Since most VIs (e.g., NDVI, EVI) are most sensitive to the canopy changes, drought monitoring based on VIs may not be useful for drought mitigation and agricultural irrigation management. SIF retrievals from satellite, comparing with optical reflectance signals, carries the information not only about the PAR absorption by chlorophyll, but also about the drought stress on plant physiology. Although previous studies used satellite-based SIF dataset for post-drought impact assessment (Lee et al., 2013; Yoshida et al., 2015; Sun et al., 2015; Wang et al., 2016), these studies did not separate the contribution of decreased $APAR_{chl}$ or deceased $SIF_{yield}$. A more recent study compared the SIF and VIs in India during a heat stress (Song et al., 2018), and found that SIF is more sensitive to heat stress than VIs. Similarly, since NDVI and EVI cannot well capture the change in chlorophyll concentration, heat stress on $APAR_{chl}$ and $SIF_{yield}$ cannot be fully separated. This study developed a new method to compare the difference between SIF signals and the reflectances, which can be applied for early drought warning at global scale. Although daily OCO-2 data has large gaps

between swaths, combining several days observation can provide enough spatial coverage considering the spatial extent for most drought events. The spatial coverage issues could be further improved using geostatistical based methods (Tadić et al., 2017), but this may need further investigation. Comparing with other meteorological drought indices, this drought monitoring technique uses only near real-time data and avoids the inter-annual anomalies caused by other factors (land cover change, crop rotation, etc.). MCD43C4 dataset uses 16 days of inputs for the model inversion, although this may lead to temporal inconsistencies for the comparison between CSIF and OCO-2 SIF, it may have limited effect due to the higher data quality during drought because of the reduced cloud coverage.

## 4.3 Cross-biome and within-biome GPP-CSIF relationship

In contrast to Sun et al. (2017), we found a large variation of GPP-CSIF relationship across sites. Compared to previous studies, our study gave higher $a_{GPP/CSIF}$ estimates, probably due to a much higher $a_{GPP/CSIF}$ value for evergreen needleleaf forest (10 out of 40 sites are ENF) (Tables S1) and slight underestimation of CSIF$_{all-daily}$ dataset. This higher $a_{GPP/CSIF}$ value for ENF was also suggested by the comparison between OCO-2 SIF and FluxCom GPP dataset (Sun et al., 2018) and other comparisons using GOSAT SIF (Guanter et al., 2012). In consistent with (Li et al., 2018b), we also found small cross-biome variation of GPP-SIF relationship. However, a large within-biome variation of $a_{GPP/CSIF}$ is also found, which contributes to a large proportion of the observed cross-sites variations rather than the cross-biome variation. Compared to studies that uses OCO-SIF within a large buffering area (e.g. 40km diameter circle in Verma et al. (2017)), we made the comparison over a much smaller area and much higher temporal frequency.

There are several explanations for the observed site-specific GPP-SIF relationship: (1) Leaf morphology may directly affect the reabsorption and scattering of SIF that leaves the foliage (Atherton et al., 2017), however, this factor is not considered in current SIF modeling (van der Tol et al., 2009; Verrelst et al., 2015) and will directly affect the model simulation of GPP-SIF relationship at the ecosystem scale (Verrelst et al., 2016; Zhang et al., 2016c). (2) Vegetation canopy characteristics also affect the reabsorption and scattering of SIF before leaving the canopy (Romero et al., 2018; Yang and van der Tol, 2018). (3) Atmospheric condition may attenuate and bias satellite SIF retrievals to some extent, but this effect is assumed to be small unless thick clouds are present (Frankenberg and Berry, 2017). (4) SIF and GPP likely have different sensitivities to environmental stresses (Flexas et al., 2002), therefore, ecosystems with frequent environmental stresses (e.g., drought) during the growing season tend to have relatively lower GPP to SIF ratio. (5) Since light saturations have less effect on SIF than GPP (Damm et al., 2015; Zhang et al., 2016c), the growing-season averaged light intensity (affected by latitude, average cloud coverage), vegetation canopy structure and leaf characteristic that relates to the light saturation will also affect GPP-SIF relationship. For example, the evergreen needleleaf forests have much higher specific leaf area and usually lower sun zenith angle, making them less prone to light saturation. These factors may vary not only across biomes, but also across sites. Therefore, within one biome type, the GPP-SIF relationship can also be different.

It is also noteworthy that clear-sky daily SIF exhibited stronger correlation with GPP (Figure 13), a possible explanation would be that the light use efficiency increases with diffused radiation, which partly compensates for the decrease in incoming PAR when cloud presents (Gu et al., 2002; Turner et al., 2006). Because satellite SIF retrieval algorithm discarded observations that were affected by thick cloud (Sun et al., 2018), the SIF retrievals from OCO-2 is positively biased than the actual SIF emission of the plants. However, during periods when thick cloud are present, the LUE also increases and so does the GPP/SIF ratio. The positive SIF retrieval biases compensated the increase in GPP/SIF ratio, and therefore, contributed to a stronger correlation between satellite retrieved SIF (rather than the actual SIF emission) and GPP.

## 4.4 Uncertainties and caveats

Although our $CSIF_{clear-inst}$ showed good performance as supported by the comparison with the clear-sky instantaneous SIF retrievals from OCO-2, the $CSIF_{all-daily}$ exhibits a slight underestimation. A possible explanation is that most SIF retrievals during overcast conditions did not pass the quality checks, such that OCO-2 SIF are more likely obtained during clear-sky conditions. This is supported by the fact that if we compare OCO-2 SIF with clear-daily SIF, the $R^2$ is even higher (Figure 6).

The canopy structure and sun-sensor geometry were not explicitly considered in our modeling and only implicitly embedded in the machine learning retrieval. Several recent studies suggest that canopy structure will affect the PAR absorption and re-absorption of SIF before leaving the canopy ($f_{esc}$ in equation 3) (Knyazikhin et al., 2013; Liu et al., 2016; Yang and van der Tol, 2018), and further affect the GPP-SIF relationship (He et al., 2017; Migliavacca et al., 2017; Zhang et al., 2016c). However, most of these studies made assumptions requiring either a dense canopy or non-reflecting soil and thus cannot be easily applied at the global scale. In addition, OCO-2 SIF data used in this study are from nadir observations, while both the MODIS and GOME-2 sensors acquire images both nadir and near nadir. Such discrepancy in observation angles may induce bidirectional effects. Since CSIF is trained based on the satellite observed SIF instead of the canopy SIF emission, and as previously discussed, it did not consider the atmospheric attenuation of SIF signal in the presence of clouds. The CSIF values are expected to be closer to the canopy SIF emission than the satellite-observed SIF at top-of-atmosphere.

The BESS PAR 4-day dataset has high overall accuracy (RRMSE=15.2%) and very little bias (1.4%). For different climate zones, the uncertainties are typically under 20%. These uncertainties do not affect the $CSIF_{clear-inst}$ data but will propagate to $CSIF_{all-daily}$.

## 5 Conclusion

In this study, using the surface reflectance from the MODIS instrument and a NN algorithm, we developed two spatially contiguous and high temporal resolution SIF datasets (CSIF). These two SIF products not only show high accuracy when

validated against the satellite retrieved OCO-2 SIF, but also exhibit reasonably high consistency with both reconstructed and satellite retrieved GOME-2 SIF. CSIF$_{all-daily}$ exhibits an increasing trend globally during 2001-2016, which is attributed to the Earth greening and not to changes in PAR. Since the CSIF dataset include most information of PAR absorption of chlorophyll, the difference between OCO-2 SIF and CSIF mostly contains the information of physiological stress on fluorescence yield.

This indicator is found to be effective for early drought warning than vegetation indices. By comparing CSIF$_{all-daily}$ with GPP estimates across 40 EC flux tower sites, the GPP-SIF relationship is found to vary across sites, and a large proportion of this comes from within-biome variation. However, this finding still requires further examination using SIF from both new satellites instruments (e.g., TROPOMI) and ground-based measurements. The high resolution CSIF dataset can be further used for regional to global carbon and water fluxes analysis.

**Code availability:**

The code used to generate the CSIF dataset is available at https://github.com/zhangyaonju/continous_SIF.

**Data availability:**

The CSIF dataset (CSIF$_{clear-inst}$, CSIF$_{clear-daily}$ and CSIF$_{all-daily}$) with a 0.5-degree spatial resolution and 4-day temporal resolution can be access through Figshare: DOI:10.6084/m9.figshare.6387494. The 0.05-degree 4-day dataset can be obtained upon request, given the large size. The MCD43C4 dataset can be access through NASA EARTHDATA (https://earthdata.nasa.gov). The BESS PAR product can be access through Environmental Ecology Lab at Seoul National University

(http://environment.snu.ac.kr/bess_rad/).

**Appendix**

**A1. Calculation of clear-sky radiation**

We calculated the clear-sky radiation following previous studies (Duffie and Beckman, 2013; Ryu et al., 2018). The total

surface shortwave radiation $R_T$ is the summation of direct surface beam radiation ($R_{sb}$) and diffused radiation ($R_{sd}$):

$$R_T = R_{sb} + R_{sd} \quad (A1)$$

$R_{sb}$ and $R_{sd}$ are calculated as the product of the top of atmosphere shortwave radiation ($R_{TOA}$) and the atmospheric transmittance for beam radiation ($\tau_b$) and that for diffused radiation ($\tau_d$):

$$R_{sb} = R_{TOA} \times \tau_b \quad (A2)$$

$$R_{sd} = R_{TOA} \times \tau_d \quad \text{(A3)}$$

where $R_{TOA}$ is calculate as a function of solar constant ($S_0$=1360.8 W m$^{-2}$), the proportion of solar irradiance within shortwave range ($\alpha$=0.98), the day of year ($n$) and the cosine of the solar zenith angle ($\cos\theta_s$):

$$R_{TOA} = S_0 \times \alpha \times \left[1 + 0.033\cos\left(\frac{2\pi n}{365}\right)\right] \times \cos\theta_s \quad \text{(A4)}$$

And $\tau_b$ is calculated as:

$$\tau_b = a_0 + a_1\exp\left(\frac{-k}{\cos\theta_s}\right) \quad \text{(A5)}$$

where $a_0$, $a_1$, and $k$ are coefficients that consider the atmospheric attenuation based on the atmosphere path length and abundance of the gases or particles that need to be adjusted for elevation:

$$a_0 = 0.4237 - 0.00821(6 - A)^2 \quad \text{(A6a)}$$

$$a_1 = 0.5055 + 0.00595(6.5 - A)^2 \quad \text{(A6b)}$$

$$k = 0.2711 + 0.01858(2.5 - A)^2 \quad \text{(A6c)}$$

where $A$ is the elevation in kilometers. The ETOPO1 Global Relief Model was used to provide the elevation information. This dataset was downloaded from National Oceanic and Atmospheric Administration (https://data.nodc.noaa.gov/cgi-bin/iso?id=gov.noaa.ngdc.mgg.dem:316) and aggregated to 0.05 degree. In this study, we did not consider the variation of these parameters for different climate and latitudinal zones since those effects are less important. The transmittance for diffused radiation ($\tau_d$) is calculated as a function of $\tau_b$:

$$\tau_d = 0.271 - 0.294\tau_b \quad \text{(A7)}$$

**Author contribution**

Y.Z. and P.G. designed the study. P.G. provided the Neuron Network training code. Y.Z. performed the analysis. P.G., S.H.A. and J.J. helped interpret the results. Y.Z. led the writing, with the input from all other authors. All authors discussed and commented on the results and the manuscript.

**Competing interests**

The authors declare no competing interests.

## Acknowledgements

This work used eddy covariance data acquired and shared by the FLUXNET community, including these networks: AmeriFlux, AfriFlux, AsiaFlux, CarboAfrica, CarboEuropeIP, CarboItaly, CarboMont, ChinaFlux, Fluxnet-Canada, GreenGrass, ICOS, KoFlux, LBA, NECC, OzFlux-TERN, TCOS-Siberia, and USCCC. The ERA-Interim reanalysis data are provided by ECMWF and processed by LSCE. The FLUXNET eddy covariance data processing and harmonization was carried out by the European Fluxes Database Cluster, AmeriFlux Management Project, and Fluxdata project of FLUXNET, with the support of CDIAC and ICOS Ecosystem Thematic Center, and the OzFlux, ChinaFlux and AsiaFlux offices. The authors would like to thank Dr. Youngryel Ryu from Seoul National University for providing the BESS PAR dataset. The authors acknowledge funding from NASA NNH16ZDA001N-AIST, Computational Technologies: "An Assessment of Hybrid Quantum Annealing Approaches for Inferring and Assimilating Satellite Surface Flux Data into Global Land Surface Models.", as well as funding from the STR3S project supported by the Belgium Space Agency.

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

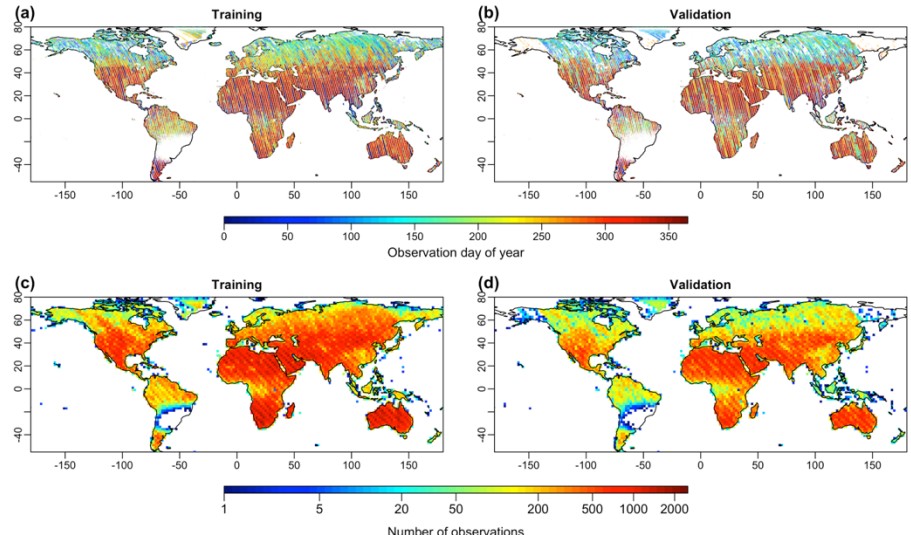

**Figure 1. Samples that were used for NN training (year 2015 and 2016) and validation (2014 and 2017). Upper panel shows the spatial distribution of observation day of year (DOY) and the bottom panel shows the spatial distribution of the sample density. Each point in (a,c) represents a 0.05-degree training gridcell. Limited observations in South America were caused by the South Atlantic Anomaly (Sun et al., 2018).**

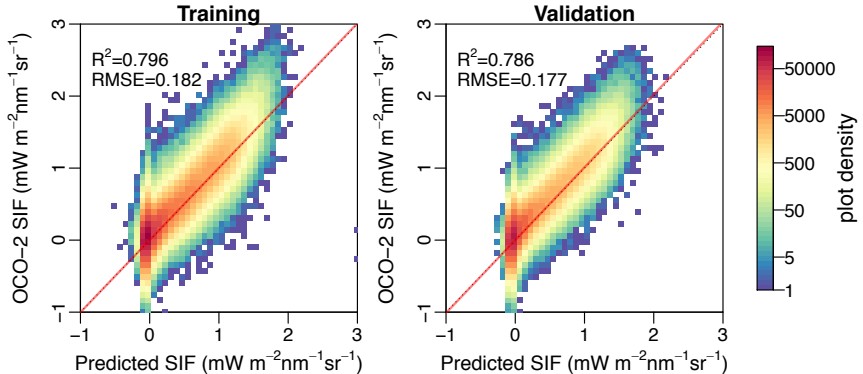

**Figure 2. Predicted SIF in comparison with the OCO-2 SIF. Red lines represent the regression slope and the black dotted lines represent the 1:1 line.**

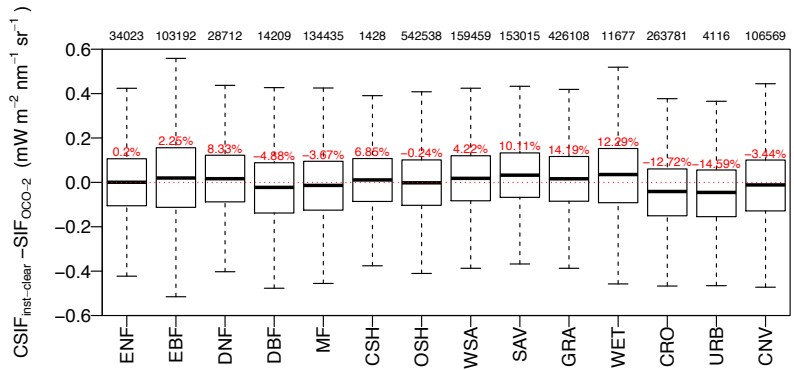

**Figure 3. Difference between CSIF$_{\text{clear-inst}}$ with SIF$_{\text{OCO-2}}$ for major biome types during 2014-2017. The MODIS land cover dataset for 2010 was used to identify the land cover type for each 0.05° grid (Friedl et al., 2010). The red percentages above each box represent the mean relative error, and the numbers on top of the figure frame represent the total sample numbers for each biome type. Abbreviations: ENF, evergreen needleleaf forest; EBF, evergreen broadleaf forest; DNF, deciduous needleleaf forest; DBF, deciduous broadleaf forest; MF, mixed forest; CSH, closed shrubland; OSH, open shrubland; WSA, woody savannas; SAV, savannas; GRA, grassland; WET, wetland; CRO, cropland; URB, urban; CNV, cropland or natural vegetation mosaics.**

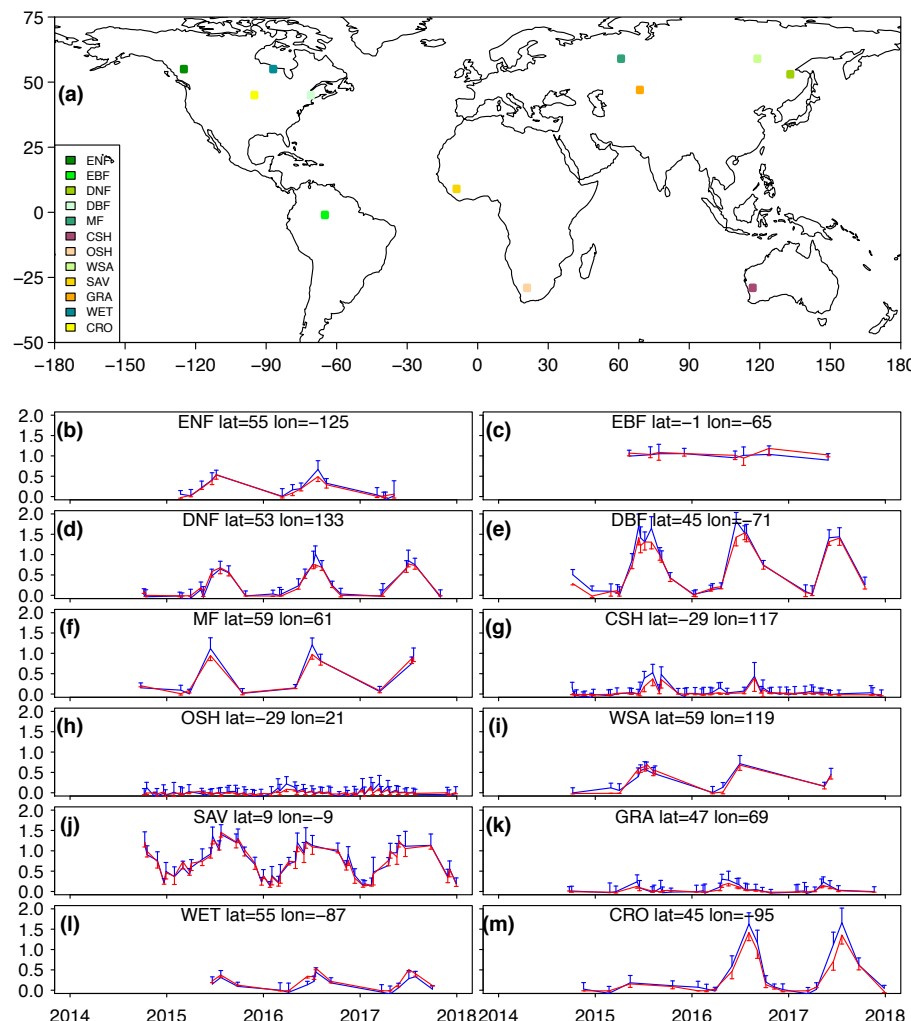

**Figure 4. Comparison of predicted SIF by NN and OCO-2 observed SIF for 12 samples (2° ×2°) of major vegetated land cover types during 2014 to 2017. All samples in the training and validation are used. The blue color represents the observed SIF by OCO-2 and the red color represent the SIF prediction by NN. The error bars represent the standard deviation of all 0.05° ×0.05° samples used to generate the 2° ×2° gridboxes. MODIS MOD12C1 V6 land cover dataset is used to select these sample gridboxes.**

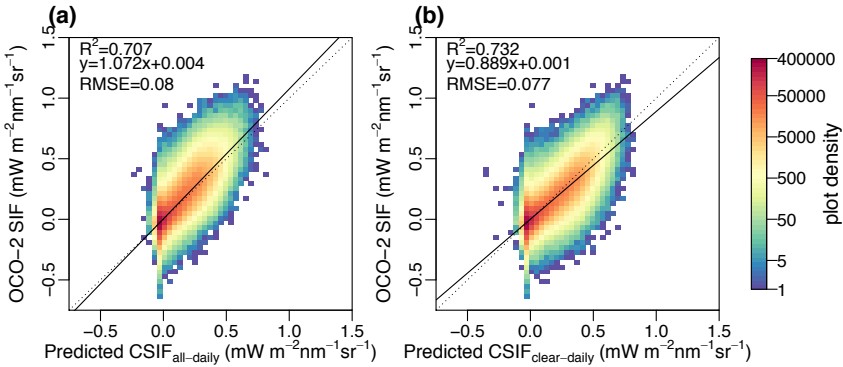

**Figure 5. Comparison between the retrieved SIF and the (a) predicted all-sky daily CSIF and (b) clear-sky daily CSIF. The instantaneous SIF retrievals from OCO-2 were converted to daily average values for comparison.**

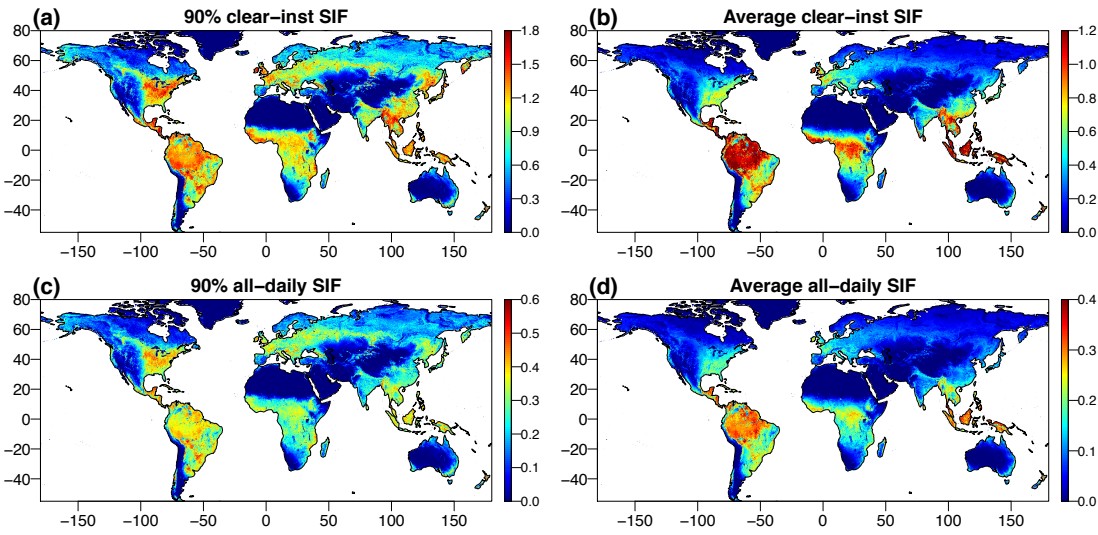

5    **Figure 6. Spatial pattern of maximum (90 percentile) and average daily values for instantaneous clear-sky SIF and all-sky daily SIF. All values are in the unit of mW m$^{-2}$ nm$^{-1}$ sr$^{-1}$.**

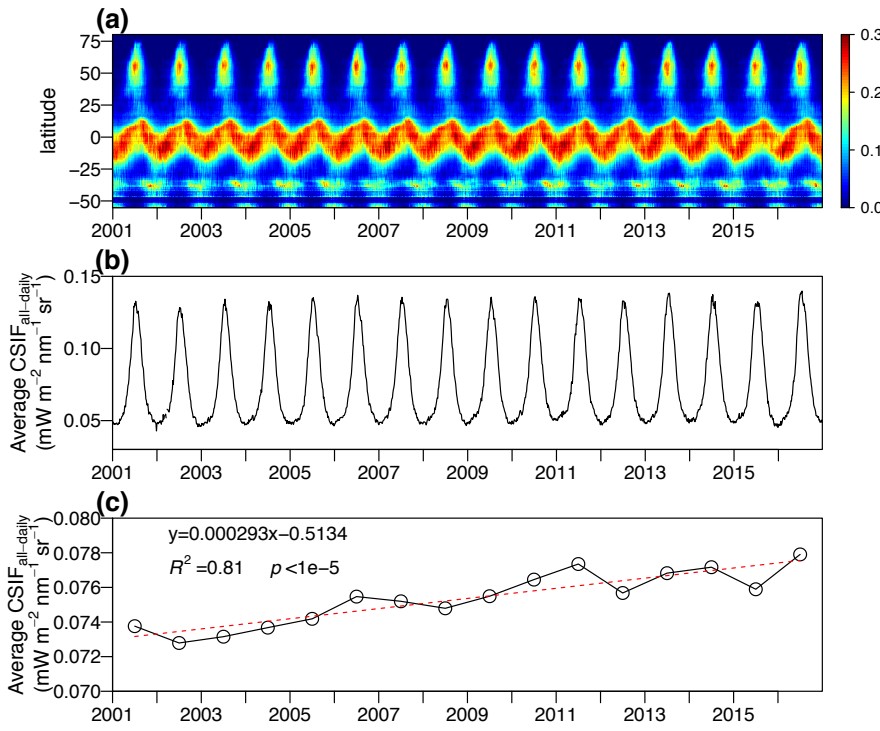

**Figure 7. Seasonal and inter-annual variation of all-sky condition daily CSIF (CSIF$_{all-daily}$). (a) the latitudinal averages of CSIF$_{all-daily}$ for each 4-day (in mW m$^{-2}$ nm$^{-1}$ sr$^{-1}$). (b) global average of CSIF$_{all-daily}$ for each 4-day. (c) the annual average CSIF$_{all-daily}$ between 2001 to 2016 (black line) with linear fit (red dashed line).**

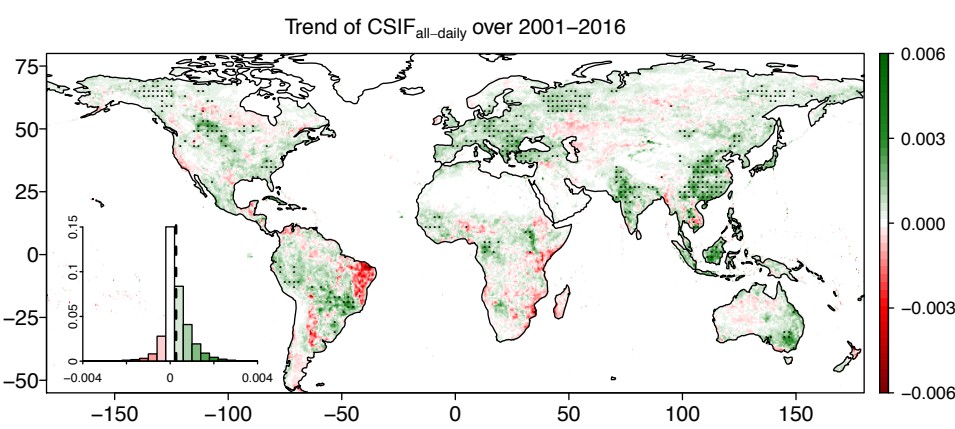

**Figure 8. Trend of annual average CSIF$_{all-daily}$ during 2003-2016. The trend is calculated by the Sen's Slope estimator. Dots represent the trend is significant (p<0.05) through a Mann-Kendall test. Inset in bottom left shows the histogram of the CSIF$_{all-daily}$ trend. Dashed vertical line represents the average trend. Barren areas with an annual average CSIF$_{all-daily}$ smaller than 0.006 mW m$^{-2}$ nm$^{-1}$ sr$^{-1}$ are screened from analysis. Trends are in the units of mW m$^{-1}$ nm$^{-1}$ sr$^{-1}$ yr$^{-1}$.**

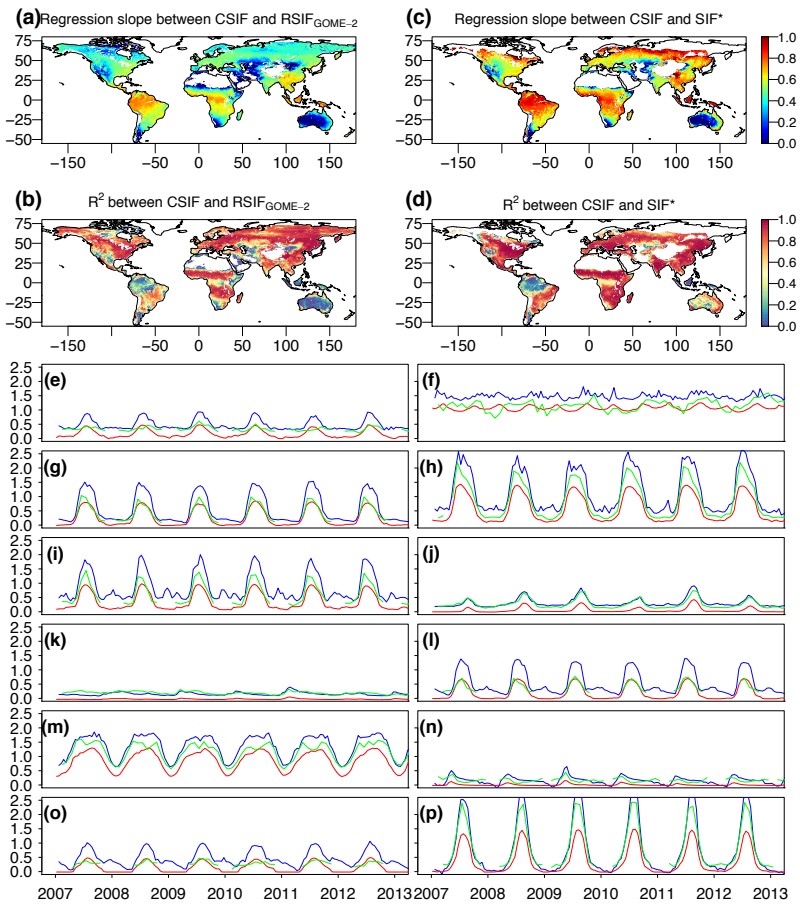

**Figure 9. Comparison between CSIF, RSIF$_{GOME-2}$ and SIF* dataset. Regression slopes and coefficient of determination (R$^2$) between the contiguous clear-sky condition instantaneous SIF from OCO-2 (CSIF$_{inst-clear}$) and the reconstructed SIF from GOME-2 (RSIF$_{GOME-2}$ a, b) or SIF* (c, d) dataset. The regressions are forced to pass the origin. The CSIF$_{clear-inst}$ is aggregated to semi-monthly and 0.5° × 0.5° spatial resolution to be consistent with RSIF$_{GOME-2}$. Comparison uses the data between 2007 to 2016 (RSIF) or 2007 to 2013 (SIF*). White regions are barren regions. (e-p) Time series comparison among CSIF (red), RSIF$_{GOME-2}$ (blue) and SIF* (green) for pixels in 12 major land cover types shown in Figure 4.**

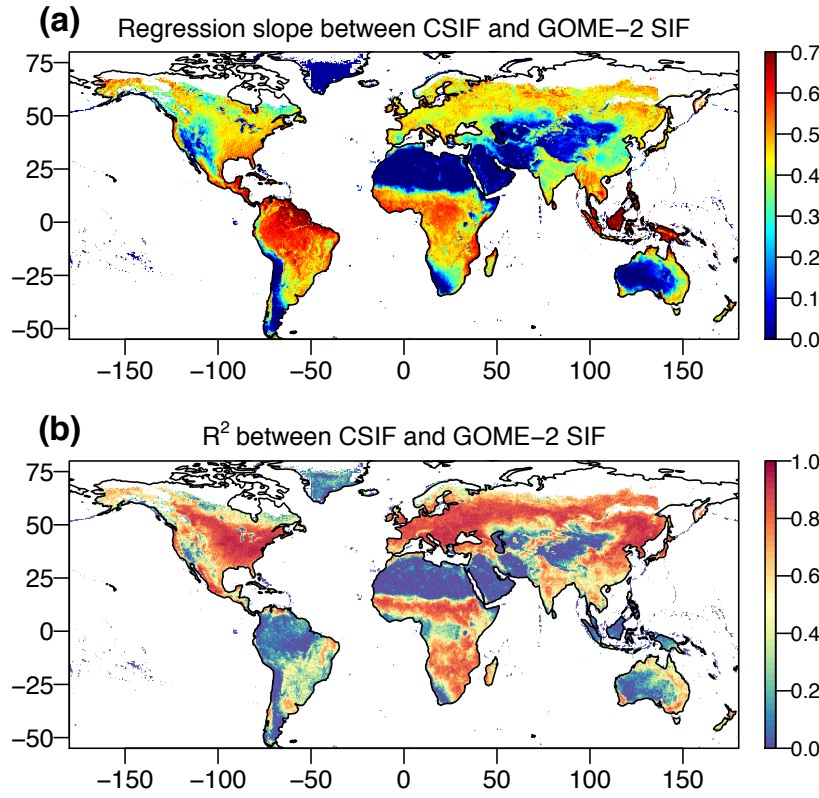

**Figure 10. Regression slopes and coefficient of determination ($R^2$) between the contiguous all-sky condition daily SIF from OCO-2 (CSIF$_{\text{all-daily}}$) and the satellite-retrieved daily SIF from GOME-2 (SIF$_{\text{GOME-2}}$). The regressions are forced to pass through the origin. The CSIF$_{\text{all-daily}}$ is aggregated to monthly and $0.5° \times 0.5°$ spatial resolution to be consistent with SIF$_{\text{GOME-2}}$. Comparison uses the data between 2007 to 2016.**

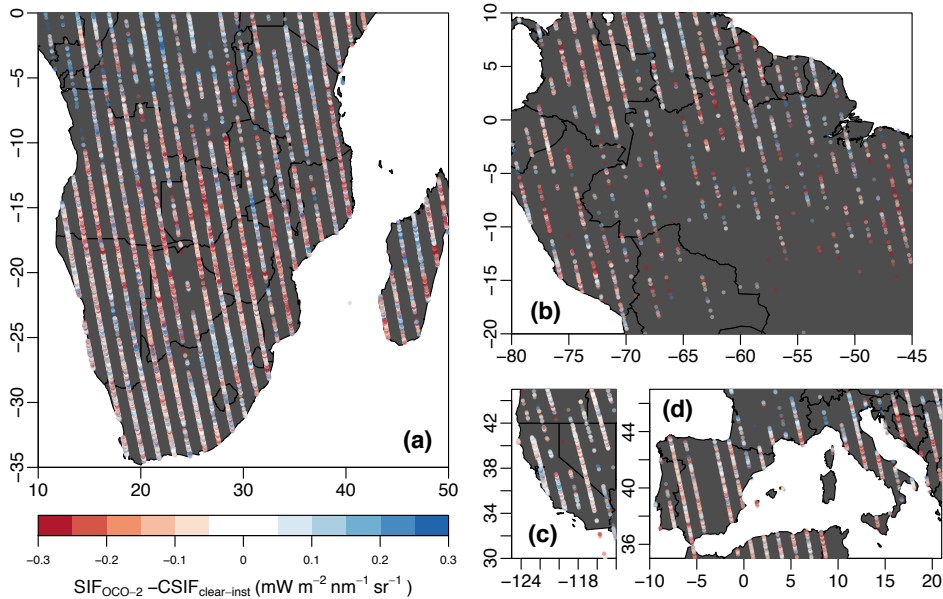

**Figure 11. Difference between the OCO-2 SIF and CSIF_clear-inst for 4 specific drought events during 2014–2017. (a) Southern Africa drought between October, 2015 and February, 2016. (b) Northeast Amazon drought between January and March, 2016. (c) California drought between January and March, 2015. (d) Southern Europe drought between July and August, 2017.**

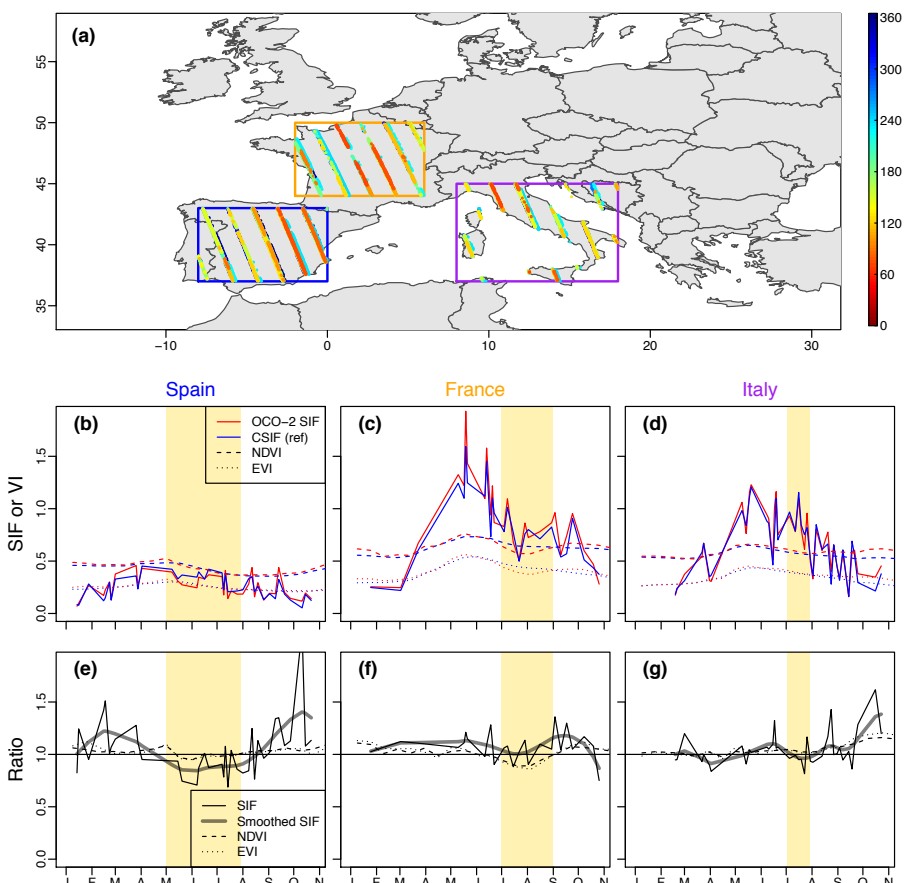

**Figure 12. (a) Spatial distribution of OCO-2 SIF observations during January 1st to November 1st in 2015. Different colours represent the observation day of year (DOY). (b-d) average OCO-2 SIF, CSIF NDVI and EVI for the three countries as indicated by three boxes in (a). For two vegetation indices, red colour represents the observations in 2015 and blue colour represents multi-year average (2000-2014). (e-g) the ratio between OCO-2 SIF and CSIF (SIF) or vegetation indices in 2015 and multi-year average. Thick grey line presents the splines smoothed SIF ratio.**

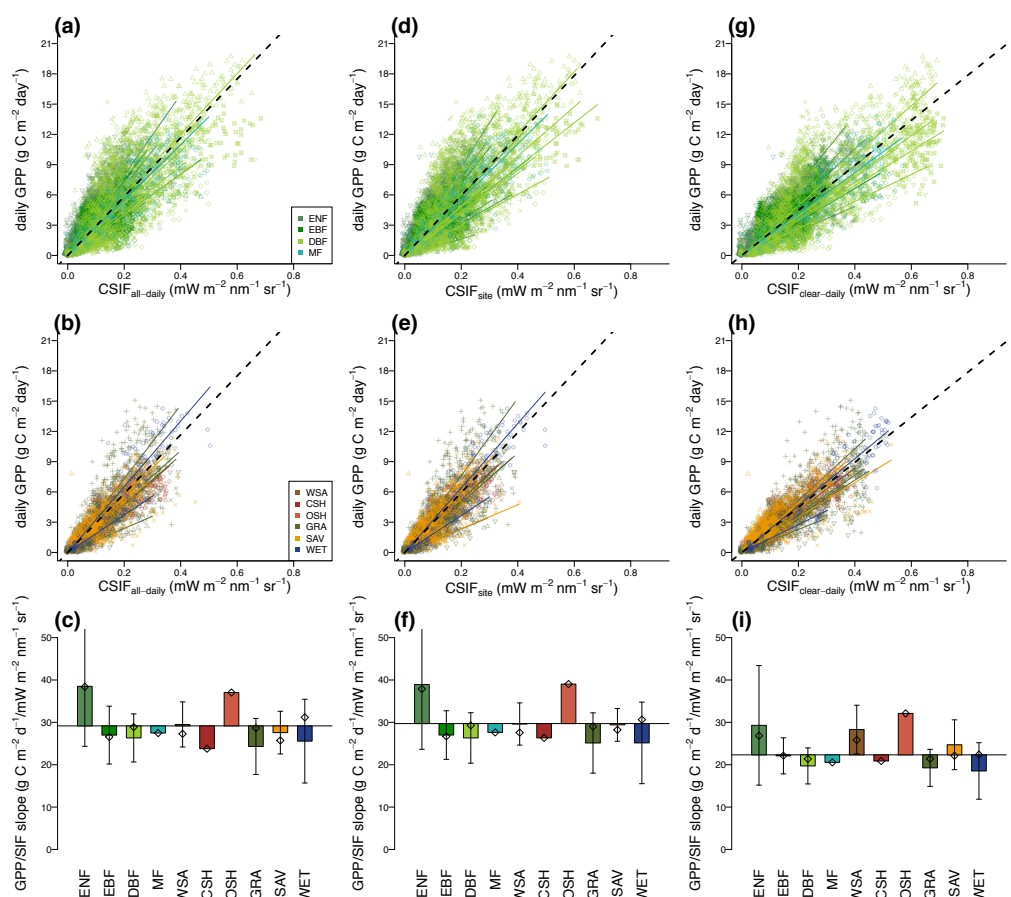

**Figure 13. Comparison between GPP estimates from 40 EC flux towers and CSIF$_{all-daily}$ (a-c) that uses BESS PAR, CSIF$_{site}$ (d-f) that uses site-measured radiation and CSIF$_{clear-daily}$ (g-i) that assume clear-sky condition. The 40 sites were grouped into forest (a,d,g) and non-forests (b,e,h). Colours-symbols combinations represent different sites. Summary of the regression slopes between GPP and CSIF for different land cover types (c,f,i). The baseline (dashed black lines) was calculated using all samples (29.71 for CSIF$_{all-daily}$ 29.18 for CSIF$_{site}$ and 22.33 for CSIF$_{clear-daily}$). Error bars represent the standard deviation of slopes across sites within this biome type. Rhombuses represent regression for each biome type when data from all sites were combined.**

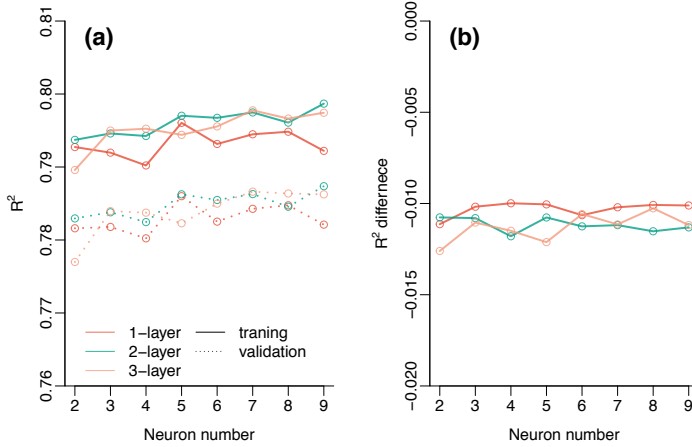

**Figure A1. (a) Comparison of model performance (R$^2$) during training and validation with a variety of NN layers (1-3) and neuron numbers for each layer (1-8). (b) difference of model performance between the training and validation for different layer and neuron combinations.**

