# Peer review of "A global spatially Contiguous Solar Induced Fluorescence (CSIF) dataset using neural networks"

_Biogeosciences, 2018_

## Referee Comment (RC1) · Anonymous Referee #1 · 24 Jul 2018

Review for "A global spatially Continuous Solar Induced Fluorescence (CSIF) dataset using neural networks" by Yao Zhang et al.

In this work the authors produce three datasets of CSIF by filling spatial and temporal gaps of SIF soundings by OCO2 using MODIS surface reflectances and machine learning. The resulting datasets in 0.05deg and 4-day resolution represent gap-filled instantaneous SIF under cloud-free conditions, cloud-free SIF integrated to a daily value and daily SIF under all-sky conditions. To illustrate the advantages and the usefulness of these high-resolution datasets they compare to another downscaled fluorescence product (RSIF, based on GOME2 and different MODIS reflectance datasets), GOME2 SIF, EC GPP and OCO2-SIF itself based on drought occurrences.

The authors convincingly argue why a new down-scaled SIF product - based on OCO2

as a new factor – is needed. At several points I do see, however, need for clarification (where the authors partly contradict themselves in my opinion), further discussion and analysis.

The main points are:

1) Does CSIF represent SIF or APARgreen (p. 3 l. 30)? At several occasions in the manuscript (e.g. p.8 l.14, p.11 l.26) the authors state that based on the reflectances SIFyield cannot be reproduced by NN. They base the drought event analysis on this assumption by comparing CSIF to OCO2 SIF (p.10 l.5). At other moments, however, they stress the close relationship of CSIF to SIF (p.8 l. 18, p.11 l. 31) and call it CSIF.

2) Related to point 1, why not compare CSIF to other estimates of APAR? Greenness index * PAR?

3) Temporal resolution: The final data sets are claimed to have 4day temporal resolution. What is left unmentioned in the manuscript is that the MCD43C4 reflectance data have daily sampling, but each value for a given day still represents a 16-day period (weighted to the central date). So the 4-daily temporal resolution might in reality represent periods of 19days length and potentially affect the downscaling and all comparisons. Please consider this in your evaluations and discussion.

4) Although data processing is described in detail, at points clarification is necessary (e.g. regarding temporal aggregation for training, quality filters of reflectance and EC GPP data, see below).

5) Spatial splitting for training-validation in addition to the temporal one, as extrapolation is not only done in time but also in space. I put it here as a recommendation.

6) I miss plots of time series throughout the manuscript both regarding training/validation as well as comparisons to other datasets. Such plots could contribute to supporting CSIF as a very useful dataset.

7) Related to point 2: An interesting comparison next to the one to RSIF and GOME2

SIF v27 would also be the one to SIF* by Duveiller and Cescatti 2016 that you cite several times.

ad 4)

a) Which period exactly do the OCO2 SIF data cover? It was launched only in September 2014, so only a few months of this year are available. Do you use the full year of 2017?

b) Aggregation of OCO2 SIF (p.4 l. 29-p.5 l.18): How do you aggregate in time? Daily, 4-daily?

c) Figure 1 nicely shows the spatial distribution of the training and validation data. What does the temporal distribution and representativeness look like? A lat-time plot might be useful here. Why are there no data for validation in Alaska and eastern Siberia?

d) You might consider removing barren areas in Sahara and central Asia from the analysis to potentially obtain clearer signals as many data points are obtained from these areas (Fig.1) and might affect the relationships in Fig.2-4.

e) p.5 ll. 12-17: If you first aggregate several soundings to 0.05deg and only afterwards integrate to a daily value, which SZA of the measurement do you use?

f) p.5 l. 21 and at several other occasions in the manuscript: which is the period covered? 2000-2017 or 2001-2016 (abstract) for CSIF?

g) Processing of reflectance data: Do you apply any quality filters? You mention the 'best atmospheric conditions' (p. 6 l. 7), what did you do? I am wondering how can you obtain 'more realistic prediction of SIF during winter' if the reflectances do not represent vegetation but snow? MCD43C4 is sampled daily but values represent 16 days worth of data.

h) You might consider adding a few sentences on BESS PAR in addition to citing Ryu et al. 2018. The quality of BESS PAR is not discussed in the discussion part.

i) In the comparisons to GOME2 SIF v27 and RSIF, is there any accounting for over-pass time and wavelength necessary between GOME2 and OCO2? A comparison to SIF* by Duveiller and Cescatti would complete the suite of products.

j) EC GPP: Please add a bit more information: Did you use FLUXNET 2015 or the LaThuile data set? Is there any reason for choosing nighttime partitioning? l. 28-30: I would think you retain those data?

Training and validation (p. 8 ll. 1-26):

-Time series of the CSIF during validation would be nice to see, with validation points overlaid, for example to illustrate the point of SIFyield in SV and GRA.

-Also crops are strongly biased.

-l.17-19: At other points it is argued that CSIF cannot accurately reproduce SIFyield effects based only on reflectances. Please clarify and discuss.

Drought monitoring application:

-Please consider the different temporal information in instantaneous OCO2 SIF versus CSIF based on 19 days worth of data and take it into account and/or discuss it.

-It would also be interesting to see here a comparison (although I see that the different temporal resolutions would mean more work) also to an estimate of APAR, RSIF, OCO2 SIF/ APAR and SIF* by Duveiller & Cescatti 2016.

A few questions when reading the Discussion:

4.1. What is the advantage of CSIF compared to using vegetation index * PAR?

Is there a trend in RSIF?

4.2. Although I fully agree, vegetation indices might not be completely blind. Sims et al. 2006 (RSE,

10.1016/j.rse.2005.01.020) argue that for some ecosystems vegetation indices can

contain information on photosynthetic yields.

4.3. In far-red SIF reabsorption should play a very minor role compared to scattering.

I am afraid I do not understand the last paragraph (specifically p.14 ll. 12-15). Please rephrase.

Minor:

P.6 l. 3: In the end four bands are used, not seven.

Fig.6a) There are also low values in 2007 and 2010 and high values in 2011 compared to 2010 and 2012 during La Nina in Australia at 40S.

Fig.9 Maybe exclude SAA from the plot as GOME2 v27 will be affected by it.

p.10 l. 27: somehow there are too many numbers.

Fig. 10: I recommend a clearer white for the zero values, the gray is difficult to distinguish from the red and blue for small values.

Discussion of uncertainties in BESS PAR is missing.

Conclusion:

MCD43C4 is based not only on Aqua.

p. 15 l. 19: 0.5 deg or 0.05deg?

p. 15 l. 27: beam radiation

subscripts sb and sd are sometimes incorrect/the same

For me, calling the SIF multiplied by a daily correction factor ' daily average SIF' is confusing. It is rather an integration over the day, 'daily integrated SIF'.

Are there any applications or areas for which you would not recommend the use of CSIF?

---

## Referee Comment (RC2) · Anonymous Referee #2 · 9 Aug 2018

In this study authors demonstrate the possibility of generating contiguous, high resolution estimates of SIF utilizing machine leaning, using as inputs sparse available OCO-2 SIF retrievals and ancillary satellite data (MODIS). Authors provided extensive error statistics and demonstrated applicability of the new approach in identifying/studying effects of drought. The study is well written and suffers only from minor issues.

Page 1, Line 10-11: "However, several issues, including low spatial and temporal resolution of the gridded datasets and high uncertainty of the individual retrievals, limit the applications of SIF.

Reviewer: Binned/averaged datasets are not the only option, there is entire family of products based on geostatistics/kriging (i.e. Tadic et al, 2017), so it could be nice to compare weaknesses/advantages to those products as well.

[Figure]

Page 1, Lines 14-15: "….we generated two global spatially continuous SIF (CSIF) datasets at moderate spatio-temporal resolutions (0.05 degree 4-day)…"

Reviewer:How did you choose the ST resolution? Why 4-day? Is it based on the expected decoorelation lenght (variability) in time? Why not 1*day?

Page 3, Lines 24-26: "In addition, OCO-2 can only generate a gridded monthly dataset at relatively coarse spatial resolution, typically at $1° \times 1°$, which limits its application in small regions. "

Reviewer: This is not quite correct. It is correct only if we limit our approach to binning/averaging, and ignore spatial autocorrelations. However, if we take autocorrelations into account, we get have estimates at much higher spatio-temporal resolutions (see Tadic et al, 2015, doi:10.5194/gmd-8-3311-2015
and Tadic et al, 2017, doi:10.5194/gmd-10-709-2017)

Page 4, Line 17:" In this study, we aim to generate a global continuous SIF (CSIF) product…"

Reviewer: Here and throughout the text, perhaps better choice of word would be contiguous. Continuous implies that there is an infinite number of estimation locations, while in practice your estimation interval is determined by the granularity of input data, in this case MODIS retrievals.

Page 4, Line 32:" The reasons for using this resolution include: (1) it is directly comparable to the OCO-2 SIF footprint s

Reviewer: This statememnt is questionable, 5x5 gives 25km2 footprint, and OCO-2 footprint is less than 3km2 in size. 8 times difference might or might not be viewed as significant.

Page 5, Lines 3-4:" assuming independent estimates and homogeneous SIF value within each gridcell…"

Reviewer: Here an additional assumption is required - the SIF has to be not only homogeneous (spatial dimension) but also constant in time.

Page 6, Line 1:" For prediction, we first aggregated the daily reflectance to 4 days. "

Reviewer:Why 4?

Page 6, Lines 10-11:" A feedforward neural network (NN) is a number of computational nodes (called neurons) structured in a multi-layer architecture."

Reviewer:In principle, NN can be a single layer structure.

Page 6, Line 16:" The rectified linear unit (ReLU) was used as the activation "

Reviewer:Is there any particular reason for this choice?

Page 7, Lines 14-15:" RSIFGOME-2 (Gentine and Alemohammad, 2018a) uses a similar machine learning technique approach to CSIF but the 15 training is based on the bi-weekly gridded SIF product from GOME-2, and 8-day MYD09A1 reflectance dataset. "

Reviewer: This choice is surprising, as GOME-2 Level 3 products cn be obtained at much higher temporal resolutions, even daily, like it was demonstrated at Tadic et al., 2017. In this case, an unnecessary degradation of information content is induced, as temporal SIF variations during biweekly periods are converted into noise. Given large footprint on GOME-2 retrievals, ML processing here played the role of the downscaling as well, which itself is a challenging process.

Page 8, Lines 1-6

Reviewer: More details are needed here, for the sake of reproducibility. Did you use regularization? What kind of regularization? L2, dropout, their parametrizatuon? How many epochs? Did you use test/validation sets approach or only test?

Page 10. Line 8:" Figure 10 shows the difference between instantaneous clear-day

OCO-2 SIF and CSIFclear-inst. "

Reviewer:Using contiguous Level 3 products based on OCO-2 data and spatio-temporal kriging would yield a figure equivalent to Fig 10, but contiguous.

---

## Author Comment (AC1) · 24 Aug 2018

Review for "A global spatially Continuous Solar Induced Fluorescence (CSIF) dataset using neural networks" by Yao Zhang et al.

In this work the authors produce three datasets of CSIF by filling spatial and temporal gaps of SIF soundings by OCO2 using MODIS surface reflectances and machine learning. The resulting datasets in 0.05deg and 4-day resolution represent gap-filled instantaneous SIF under cloud-free conditions, cloud-free SIF integrated to a daily value and daily SIF under all-sky conditions. To illustrate the advantages and the usefulness of these high-resolution datasets they compare to another downscaled
fluorescence product (RSIF, based on GOME2 and different MODIS reflectance datasets), GOME2 SIF, EC GPP and OCO2-SIF itself based on drought occurrences.

The authors convincingly argue why a new down-scaled SIF product - based on OCO2 as a new factor – is needed. At several points I do see, however, need for clarification (where the authors partly contradict themselves in my opinion), further discussion and analysis.

Response: Thanks for your nice summary. We address your concerns point-by-point below.

The main points are:

1) Does CSIF represent SIF or APARgreen (p. 3 l. 30)? At several occasions in the manuscript (e.g. p.8 l.14, p.11 l.26) the authors state that based on the reflectances SIFyield cannot be reproduced by NN. They base the drought event analysis on this assumption by comparing CSIF to OCO2 SIF (p.10 l.5). At
other moments, however, they stress the close relationship of CSIF to SIF (p.8 l. 18, p.11 l. 31) and call it CSIF.

Response: As we discussed in the introduction, $APAR_{chl}$ (or $APAR_{green}$) is the primary driver of the SIF variation at subdaily or seasonal scales (Du et al., 2017). The $SIF_{yield}$, on the other hand, can be relatively stable at monthly or seasonal scales although diurnal variations exist. A recent study using
canopy SIF observations at a paddy rice site also showed that SIF is strongly correlated with APAR more so than with GPP at both half-hourly ($R^2=0.82$) and daily ($R^2=0.85$) scale (Yang et al., 2018). However, this only holds when no strong environmental stress is present. When environmental stress exists, the $SIF_{yield}$ should play a more important role and decrease SIF from its normal conditions (Liu et al., 2018).

We call this dataset CSIF since it reproduces most of the SIF variations retrieved by the OCO-2 satellite. The drought may happen for a limited space and time and deviate CSIF from OCO-2 SIF, but the overall relationship between CSIF and OCO-2 SIF is still very close.

Liu, L., Yang, X., Zhou, H., Liu, S., Zhou, L., Li, X., Yang, J., Han, X. and Wu, J.: Evaluating the utility of
solar-induced chlorophyll fluorescence for drought monitoring by comparison with NDVI derived
       from wheat canopy, Science of The Total Environment, 625, 1208–1217,
       doi:10.1016/j.scitotenv.2017.12.268, 2018.
Yang, K., Ryu, Y., Dechant, B., Berry, J. A., Hwang, Y., Jiang, C., Kang, M., Kim, J., Kimm, H., Kornfeld, A.
       and Yang, X.: Sun-induced chlorophyll fluorescence is more strongly related to absorbed light than
to photosynthesis at half-hourly resolution in a rice paddy, Remote Sensing of Environment, 216,
       658–673, doi:10.1016/j.rse.2018.07.008, 2018.

2) Related to point 1, why not compare CSIF to other estimates of APAR? Greenness index * PAR?

Response: Thanks for your suggestion, the comparison between SIF and APAR has been carried out by
several other studies both at regional scale (Zhang et al., 2016) and at site level (Yang et al., 2015; Yang et al., 2018). However, accurate acquisition of $APAR_{chl}$ (rather than APAR) that drives the SIF and GPP is problematic. When using $APAR_{canopy}$ observations, the difference between $APAR_{chl}$ and $APAR_{canopy}$ (mostly related to chlorophyll concentration in the canopy) will be mistakenly interpreted as fluorescence yield or light use efficiency (Zhang et al., 2018).

Another problem with greenness index * PAR is that the greenness indices do not align with fPAR at 0. Several studies use different factors to correct for this misalignment, ranging from 0.02 (Ruimy et al., 1994) to 0.28 (Lind and Fensholt, 1999) for NDVI. In essence, this correction factor is affected by the soil (or snow) background and no universal value may be used. CSIF may be able to better capture the soil background information since it is directly trained on OCO-2 SIF, the soil background information
can be correctly picked up by the NN and the resulting CSIF can be more closely related to SIF than greenness index * PAR.

Ruimy, A., Saugier, B., & Dedieu, G. (1994). Methodology for the estimation of terrestrial net primary
       production from remotely sensed data. *Journal of Geophysical Research: Atmospheres*, *99*(D3),
       5263-5283.
Lind, M., & Fensholt, R. (1999). The spatio-temporal relationship between rainfall and vegetation
       development in Burkina Faso. *Geografisk Tidsskrift, 2*.

Yang, K., Ryu, Y., Dechant, B., Berry, J. A., Hwang, Y., Jiang, C., Kang, M., Kim, J., Kimm, H., Kornfeld, A. and Yang, X.: Sun-induced chlorophyll fluorescence is more strongly related to absorbed light than to photosynthesis at half-hourly resolution in a rice paddy, Remote Sensing of Environment, 216, 658–673, doi:10.1016/j.rse.2018.07.008, 2018.

Yang, X., Tang, J., Mustard, J. F., Lee, J., Rossini, M., Joiner, J., Munger, J. W., Kornfeld, A. and Richardson, A. D.: Solar-induced chlorophyll fluorescence that correlates with canopy photosynthesis on diurnal and seasonal scales in a temperate deciduous forest, Geophysical Research Letters, 42(8), 2977–2987, doi:10.1002/2015GL063201, 2015.

Zhang, Y., Xiao, X., Jin, C., Dong, J., Zhou, S., Wagle, P., Joiner, J., Guanter, L., Zhang, Y., Zhang, G., Qin, Y., Wang, J. and Moore, B.: Consistency between sun-induced chlorophyll fluorescence and gross primary production of vegetation in North America, Remote Sensing of Environment, 183, 154–169, doi:10.1016/j.rse.2016.05.015, 2016.

Zhang, Y., Xiao, X., Wolf, S., Wu, J., Wu, X., Gioli, B., Cescatti, A., Van Der Tol, C., Zhou, S., Gough, C., Gentine, P., Zhang, Y., Steinbrecher, R. and Ardö, J.: Spatio-temporal convergence of maximum daily light-use efficiency based on radiation absorption by canopy chlorophyll, Geophysical Research Letters, (45), 3508–3519, doi:10.1029/2017GL076354, 2018.

3) Temporal resolution: The final data sets are claimed to have 4day temporal resolution. What is left unmentioned in the manuscript is that the MCD43C4 reflectance data have daily sampling, but each value for a given day still represents a 16-day period (weighted to the central date). So the 4-daily temporal resolution might in reality represent periods of 19days length and potentially affect the downscaling and all comparisons. Please consider this in your evaluations and discussion.

Response: Thanks for your suggestion, we also realized that the MCD43C4 dataset uses 16 days of data as input and the day of interest is emphasized. This may cause potential temporal inconsistency in the dates it represents. However, we think this would have limited effect due to the following reasons: (1) the vegetation growth is continuous in time, the optical properties that will be obtained by satellite would not change abruptly. (2) although MCD43C4 uses 16-day worth of inputs, it also emphasizes the day of interest. Nevertheless, we agree this should be made clear in the revised manuscript and we added the discussions of this issue in the Method section 2.2.

4) Although data processing is described in detail, at points clarification is necessary (e.g. regarding temporal aggregation for training, quality filters of reflectance and EC GPP data, see below).

Response: We tried to resolve your concerns and clarify the data filtering issues.

5) Spatial splitting for training-validation in addition to the temporal one, as extrapolation is not only done in time but also in space. I put it here as a recommendation.

Response: Since the largest variation in vegetation activity are the spatial and seasonal variability, splitting samples by years will maximize the spatial and seasonal coverage in the training and validation samples. In addition, since we need to randomize all the samples prior training the NN, using samples from entire years tend to get more randomly distributed samples in the space-season domain. If we separate the samples by latitude, there may be ecosystems/areas that only occur in the validation but not the training dataset, and the model may be biased.

6) I miss plots of time series throughout the manuscript both regarding training/validation as well as comparisons to other datasets. Such plots could contribute to supporting CSIF as a very useful dataset.

Response: We agree that such time series plots would be helpful to get a sense of seasonal variations. We therefore added a figure for the training and validation dataset (see Figure R1 below). We also revised the figure for the RSIF and SIF* comparison and added the time-series comparison in addition to the regression analysis.

7) Related to point 2: An interesting comparison next to the one to RSIF and GOME2

SIF v27 would also be the one to SIF* by Duveiller and Cescatti 2016 that you cite several times.

Response: Thanks for your suggestions, we added this comparison in the revised manuscript and a paragraph of discussion is also added to Section 3.3.

ad 4)

a) Which period exactly do the OCO2 SIF data cover? It was launched only in Septem- ber 2014, so only a few months of this year are available. Do you use the full year of 2017?

Response: We used the data between September 2014 to December 2017. This has been clarified in the revised manuscript. It is normal to have less samples for validation than for training.

b) Aggregation of OCO2 SIF (p.4 l. 29-p.5 l.18): How do you aggregate in time? Daily, 4-daily?

Response: The aggregation is conducted for each OCO-2 SIF files (daily), since the revisit cycle of OCO-2 SIF is 16 days, and only nadir view SIF observations were used, using daily or 4-day aggregation should not change the value of the samples.

c) Figure 1 nicely shows the spatial distribution of the training and validation data. What does the temporal distribution and representativeness look like? A lat-time plot might be useful here. Why are there no data for validation in Alaska and eastern Siberia?

Response: In Figure 1 (a,b), the color of the dots represents the observations' day of year. Although these dots overlapped in the lower latitude, most of the lower latitude have training and validation samples throughout the year. The boreal regions only have samples during the summer where sun zenith angle is relatively low and no snow or ice cover.

Since the OCO-2 satellite was launched in July 2014 and starting to obtaining data after September that year, limited observations is acquired in boreal regions that year. In 2017, the satellite also experienced malfunctioning in August and early September, making limited observations in the boreal growing season.

d) You might consider removing barren areas in Sahara and central Asia from the analysis to potentially obtain clearer signals as many data points are obtained from these areas (Fig.1) and might affect the relationships in Fig.2-4.

Response: We agree that using the non-barren samples to train the dataset may yield slightly better performance. However, in this study, we aim to generate a global spatially continuous dataset. The SIF dynamic in these sparsely vegetated may be potentially of interest to future studies. Therefore, using the training samples from these regions are necessary.

e) p.5 ll. 12-17: If you first aggregate several soundings to 0.05deg and only afterwards integrate to a daily value, which SZA of the measurement do you use?

Response: If we call a 0.05-degree aggregated pixel a training sample, all the retrievals to generate this sample are from the consecutive observations within a very short period of time, usually within several seconds. The SZA during this period have little change and the average SZA from these retrievals is used as the SZA of this aggregated sample.

f) p.5 l. 21 and at several other occasions in the manuscript: which is the period covered? 2000-2017 or 2001-2016 (abstract) for CSIF?

Response: The clear-sky instantaneous/daily CSIF only requires surface reflectances and calculated SZA as input, therefore, these two datasets span from 2000 to 2017. Limited by the availability of BESS PAR (2000-2016), the all-sky daily CSIF only span from 2000-2016. We have corrected the data coverage throughout the manuscript.

g) Processing of reflectance data: Do you apply any quality filters? You mention the 'best atmospheric conditions' (p. 6 l. 7), what did you do? I am wondering how can you obtain 'more realistic prediction of SIF during winter' if the reflectances do not represent vegetation but snow? MCD43C4 is sampled daily but values represent 16 days worth of data.

Response: The MCD43C4 dataset has filtered out bad estimates (cloud affected) during the pre-processing for the model inversion (Schaaf et al., 2002), therefore no additional quality check is applied in this study. However, the MCD43C4 also suffers from spatial gaps caused by failing to implement the inversion model and predict the NBAR. To fill the gaps in the MCD43C4 dataset, we used the algorithm (Zhang et al., 2017) for each band to reconstruct the 4-day observations.

Since the RossThick/LiSparse-Reciprocal BRDF model used to generate MCD43C4 dataset may fail under contaminated atmospheric conditions, the observations obtained at bad atmospheric conditions is likely to be filter out during the model inversion and aggregation processes. We realized that this statement is not accurate and we have revised this to:

*"Since this processing does not involve any extra information and only uses the reflectance*
*observations from the successful model inversion, it should be comparable to the reflectance used for NN training."*

We deliberately included the snow affected pixels in the training and prediction, so that if the pixel is covered by snow (usually have quite different surface reflectances than vegetated surface), its SIF values can be correctly predicted. If the vegetation is covered by snow, it is likely to have minimum
APAR and little SIF, both spectral observation (reflectance) from MODIS and SIF from OCO-2 would contain little vegetation signal. This enables us to get consistent SIF values with the satellite retrievals.

We added some discussion about the temporal representation of MCD43C4 in Section 2.2

Zhang, Y., Xiao, X., Wu, X., Zhou, S., Zhang, G., Qin, Y. and Dong, J.: A global moderate resolution dataset of gross primary production of vegetation for 2000–2016, Scientific Data, 4, 170165,
doi:10.1038/sdata.2017.165, 2017.

Schaaf, C. B., Gao, F., Strahler, A. H., Lucht, W., Li, X., Tsang, T., Strugnell, N. C., Zhang, X., Jin, Y., Muller, J.-P., Lewis, P., Barnsley, M., Hobson, P., Disney, M., Roberts, G., Dunderdale, M., Doll, C., d'Entremont, R. P., Hu, B., Liang, S., Privette, J. L. and Roy, D.: First operational BRDF, albedo nadir reflectance products from MODIS, Remote Sensing of Environment, 83(1–2), 135–148,
doi:10.1016/S0034-4257(02)00091-3, 2002.

h) You might consider adding a few sentences on BESS PAR in addition to citing Ryu et al. 2018. The quality of BESS PAR is not discussed in the discussion part.

Response: We have now added several sentences in Section 4.4 to discuss the quality of BESS PAR dataset and its effect on CSIF.

i) In the comparisons to GOME2 SIF v27 and RSIF, is there any accounting for over- pass time and wavelength necessary between GOME2 and OCO2? A comparison to SIF* by Duveiller and Cescatti would complete the suite of products.

Response: We did not consider the differences of SIF emission at 757nm and 740nm. Since the regression is conducted at the temporal domain for each pixel, the temporal variation of ratio
between SIF at 757nm to SIF at 740nm is thought to be limited and does not affect the coefficient of determination. Since all the comparisons uses the daily averaged SIF, there is no need to consider the overpass time differences.

We did not add the SIF* dataset into comparison in the first version of the manuscript since it is not open to the public. In this revised version, we added the comparison between CSIF and SIF* together
with the RSIF dataset. We also added a paragraph to describe this results in Section 3.3

j) EC GPP: Please add a bit more information: Did you use FLUXNET 2015 or the LaThuile data set? Is there any reason for choosing nighttime partitioning? l. 28-30: I would think you retain those data?

Response: We used the FLUXNET2015 dataset. This is made clearer in the revised manuscript. The nighttime partitioning method is now better described and can be regarded as more independent
estimate of GPP. The daytime method assumes a hyperbolic dependence of PAR which also affect SIF. However, the difference between these two is minor (see the detailed comparison in Zhang et al., 2018).

Yes, we do mean those data are retained. This has been corrected in the revised manuscript.

Zhang, Y., Xiao, X., Zhang, Y., Wolf, S., Zhou, S., Joiner, J., Guanter, L., Verma, M., Sun, Y., Yang, X., Paul-
Limoges, E., Gough, C. M., Wohlfahrt, G., Gioli, B., van der Tol, C., Yann, N., Lund, M. and de Grandcourt, A.: On the relationship between sub-daily instantaneous and daily total gross primary production: Implications for interpreting satellite-based SIF retrievals, Remote Sensing of Environment, 205, 276–289, doi:10.1016/j.rse.2017.12.009, 2018.

Training and validation (p. 8 ll. 1-26):

-Time series of the CSIF during validation would be nice to see, with validation points overlaid, for example to illustrate the point of SIFyield in SV and GRA.

Response: We agree that the time series validation of the CSIF data would be a great complimentary to the scatter plot and the boxplot, here we select 12 2°×2° gridboxes, and plot their time-series for the SIF retrieval from OCO-2, and CSIF predicted values for the corresponding pixels and date. The CSIF
generally showed minimum differences from the original OCO-2 SIF, except slight underestimation during the peak growing season for cropland and deciduous broadleaf forest for some years. Since the SIF$_{yield}$ may only have significant effect during strong environmental stress period, which will be further shown and discussed in Section 3.4 and 4.2, we did not deliberately select samples that are affected by the drought.

[Figure]

Figure R1. Comparison of predicted SIF by NN and OCO-2 observed SIF for 12 samples (2° ×2°) of major vegetated land cover types during 2014 to 2017. All samples in the training and validation are used. The blue color represents the observed SIF by OCO-2 and the red color represent the SIF prediction by NN. The error bars represent the standard deviation of all 0.05° ×0.05° samples used to
generate the 2° ×2° gridboxes. MODIS MOD12C1 V6 land cover dataset is used to select these sample gridboxes.

-Also crops are strongly biased.

Response: Thanks for pointing out, we also described and discussed this in the revised manuscript.

-l.17-19: At other points it is argued that CSIF cannot accurately reproduce SIFyield effects based only on reflectances. Please clarify and discuss.

Response: Through a sensitivity test using the SCOPE model, the fluorescence yield variations under unstressed conditions are small, especially compare to the variation of APAR. This is also supported by a recent study carried out at a paddy rice cropland, suggesting that SIF contains most information of APAR than GPP (Yang et al., 2018). The short-term variations of fluorescence yield may be significant and become important when strong stress presents (Frankenberg et al., 2017), which is not prevalent for the training samples. The long-term (seasonal) or cross-biome changes in fluorescence yield may be related to leaf nitrogen content or carboxylation rate that may be in embedded in the surface reflectance and implicitly picked up by the Neural Network.

We highlighted the differences between the mean fluorescence yield and stress induced fluorescence yield changes in the revised manuscript.

Frankenberg, C. and Berry, J. A.: Solar Induced Chlorophyll Fluorescence: Origins, Relation to Photosynthesis and Retrieval, in Comprehensive Remote Sensing, pp. 143–162, Elsevier., 2017.

Drought monitoring application:

-Please consider the different temporal information in instantaneous OCO2 SIF versus CSIF based on 19 days worth of data and take it into account and/or discuss it.

Response: Thanks for your suggestion. Since during the drought period, the atmospheric conditions are less likely to be contaminated and the MCD43C4 dataset tends to give the best estimate using the full model inversion with the day of interest highlighted. This reduced the possibility of using observations far from the day of interest.

We agree this could be an important issue for the drought monitoring using CSIF. We therefore added some discussion in Section 4.2 about this issue.

*"MCD43C4 dataset uses 16 days of inputs for the model inversion, although this may lead to temporal inconsistency for the comparison between CSIF and OCO-2 SIF, it may have limited effect due to the higher data quality during drought with less cloud coverage."*

-It would also be interesting to see here a comparison (although I see that the different temporal resolutions would mean more work) also to an estimate of APAR, RSIF, OCO2 SIF/ APAR and SIF* by Duveiller & Cescatti 2016.

Response: Thanks for your suggestion. For the drought monitoring, as we discussed later in the Section 4.2, we would like to highlight the different basis for two drought monitoring methods, the VI based ones use interannual anomalies and detect drought only at later stage. The CSIF based method focuses more on the physiological stress on fluorescence yield. Adding additional APAR dataset would introduce uncertainties related to both PAR and fPAR (fPAR$_{chl}$ and fPAR$_{canopy}$, see our response to comments related Sims et al., 2005). In addition, the SIF* dataset does not have coverage for the period of interest.

A few questions when reading the Discussion:

4.1. What is the advantage of CSIF compared to using vegetation index * PAR?

Response: please refer to our responses to the first two comments about the difference between CSIF and vegetation index*PAR

Is there a trend in RSIF?

Response: Since RSIF starts from January 15$^{th}$ 2007, we only have 9 full years to calculate the trend for the RSIF, which is much shorter than CSIF dataset. The spatial patterns are somewhat similar, for example a positive trend in Europe, southeast China, North India, and a negative trend in East Brazil. Since RSIF is not the focus of this study and the comparison is during different period of time, we did not include this figure in the paper.

[Figure]

Trend of RSIF$_{all-daily}$ over 2008–2016

Figure R2. The trend of RSIF during 2008 to 2016. Black dots represent trend is significant at P<0.05 through the Mann-Kendall test. The trend is estimated by the Sen's slope estimator.

4.2. Although I fully agree, vegetation indices might not be completely blind. Sims et al. 2006 (RSE, 10.1016/j.rse.2005.01.020) argue that for some ecosystems vegetation indices can contain
information on photosynthetic yields.

Response: We agree that several studies including Sims et al., 2006; Wu et al., 2010 suggested that the vegetation indices (VI) have positive correlation with the light use efficiency and a VI model is further developed. However, we argue that these positive correlations may be caused by different definitions of light use efficiency (Gitelson and Gamon, 2015; Zhang et al., 2018). The fraction of light absorbed by
the canopy chlorophyll ($fPAR_{chl}$) is what drives the photosynthesis, but it is only a proportion of the total light absorbed by canopy ($fPAR_{canopy}$). There may be positive correlations between the ratio of these two FPAR definitions ($fPAR_{chl}/ fPAR_{canopy}$) and the VI, but the correlation between these two does not support the usage of VI for plant physiological stress, which is also suggested by Sims et al., 2005, as the correlation break down during drought. In other words, the VIs may correlate with the seasonal
changes of chlorophyll concentration, but the yield information is only related to the energy partitioning after absorption by photosystems by its definition, and should be independent from the canopy characteristics.

Wu, C., Niu, Z. and Gao, S.: Gross primary production estimation from MODIS data with vegetation
index and photosynthetically active radiation in maize, Journal of Geophysical Research
          Atmospheres, 115(12), 1–11, doi:10.1029/2009JD013023, 2010.
Gitelson, A. A. and Gamon, J. A.: The need for a common basis for defining light-use efficiency:
          Implications for productivity estimation, Remote Sensing of Environment, 156, 196–201,
          doi:10.1016/j.rse.2014.09.017, 2015.
Zhang, Y., Xiao, X., Wolf, S., Wu, J., Wu, X., Gioli, B., Cescatti, A., Van Der Tol, C., Zhou, S., Gough, C.,
          Gentine, P., Zhang, Y., Steinbrecher, R. and Ardö, J.: Spatio-temporal convergence of maximum
          daily light-use efficiency based on radiation absorption by canopy chlorophyll, Geophysical
          Research Letters, (45), 3508–3519, doi:10.1029/2017GL076354, 2018.

4.3. In far-red SIF reabsorption should play a very minor role compared to scattering.

Response: Thanks for your suggestion, we agree that the reabsorptions are relatively small at far-red SIF and recent studies suggest scattering is important for SIF at far-red wavelength (Yang et al., 2018). We revised these sentences to make them accurate.

I am afraid I do not understand the last paragraph (specifically p.14 ll. 12-15). Please rephrase.

Response: We rewrote this paragraph in the revised version of the manuscript. It should be clearer now.

Minor:
P.6 l. 3: In the end four bands are used, not seven.

Response: revised as suggested.

Fig.6a) There are also low values in 2007 and 2010 and high values in 2011 compared to 2010 and 2012 during La Nina in Australia at 40S.

Response: Thanks for pointing out, since the austral summer is between the two years, we revised this sentence to accurately describe the dry and wet years.

*"Low SIF values can be found in dry years (2006-2007, 2009-2010) while high values were observed in*
*wet or normal years (2010-2011, 2012-2015)."*

Fig.9 Maybe exclude SAA from the plot as GOME2 v27 will be affected by it. p.10 l. 27: somehow there are too many numbers.

Response: Thanks for your suggestions, we agree that GOME-2 SIF in parts of South America is affected by the SAA. SIF retrievals from these areas has high uncertainty, and the correlation between
CSIF and GOME-2 SIF is much lower than other regions that have similar seasonal variability. We decided to keep the comparison in this region since it is interesting when comparing with the RSIF for the same region. We added additional sentences describing this phenomenon and its causes.

We also corrected the slope range issues for the CSIF$_{site}$.

Fig. 10: I recommend a clearer white for the zero values, the gray is difficult to distinguish from the red
and blue for small values.

Response: We revised this figure as suggested. However, to make these white points distinguishable, we changed the background to dark grey.

Discussion of uncertainties in BESS PAR is missing.

Response: Thanks for pointing out, we added the discussion about BESS PAR performance in Section
4.4

Conclusion:
MCD43C4 is based not only on Aqua.

Response: We have removed Aqua in this sentence.

p. 15 l. 19: 0.5 deg or 0.05deg?

Response: The 0.5-degree dataset is provided through Figshare. The raw 0.05-degree dataset exceeded the storage limit and can be obtained upon request. We added this information in the revised text.

p. 15 l. 27: beam radiation
subscripts sb and sd are sometimes incorrect/the same

Response: two misuses of sb are corrected.

For me, calling the SIF multiplied by a daily correction factor 'daily average SIF' is confusing. It is rather an integration over the day, 'daily integrated SIF'.

Response: We followed the OCO-2 SIF user guide (https://docserver.gesdisc.eosdis.nasa.gov/public/project/OCO/OCO2_SIF_B7000_Product_Description
n_090215.pdf) and name it daily average SIF. It makes sense calling it daily average since the output represents average SIF over 24 hours, although SIF values are 0 during the night.

Are there any applications or areas for which you would not recommend the use of CSIF?

Response: As we have discussed, the CSIF dataset uses only the spectral information and is not suitable to extract plant physiological information by using it alone.

---

## Author Comment (AC2) · 24 Aug 2018

In this study authors demonstrate the possibility of generating contiguous, high resolution estimates of SIF utilizing machine leaning, using as inputs sparse available OCO- 2 SIF retrievals and ancillary
satellite data (MODIS). Authors provided extensive error statistics and demonstrated applicability of the new approach in identifying/studying effects of drought. The study is well written and suffers only from minor issues.

Response: We thank the reviewer for the clear summary and positive comments.

Page 1, Line 10-11: "However, several issues, including low spatial and temporal reso- lution of the
gridded datasets and high uncertainty of the individual retrievals, limit the applications of SIF.

Reviewer: Binned/averaged datasets are not the only option, there is entire family of products based on geostatistics/kriging (i.e. Tadic et al, 2017), so it could be nice to compare weaknesses/advantages to those products as well.

Response: Thanks for your suggestion. Indeed, geo-statistics is another option to generate high
resolution contiguous SIF dataset. However, considering that we already have quite a few contents in this study, and the comparison between different methods in generating high resolution SIF dataset can foster an independent study, we decided not adding the comparison in this paper. Yet, we mention that some other family of products could be used.

Page 1, Lines 14-15: "....we generated two global spatially continuous SIF (CSIF) datasets at moderate
spatio-temporal resolutions (0.05 degree 4-day). . ."

Reviewer: How did you choose the ST resolution? Why 4-day? Is it based on the expected decoorelation lenght (variability) in time? Why not 1*day?

Response: The spatial resolution of 0.05 is fine enough for comparison with site level observations (compared with swath based OCO-2 SIF), it also provides global coverage without dividing the Earth
surface into tiles (like the 500m MODIS dataset), which simplifies the global application. In addition, many input datasets are available at the 0.05 degree (BESS PAR for example).

We chose this temporal resolution to reach a balance among applications requirements, information redundancy, and dataset sizes. For most GPP dataset, they are either 8-day or monthly temporal
resolution. For widely used SIF datasets, GOME-2 for example, they are mostly at the monthly or semi-monthly resolution. Since SIF is often used as a reliable proxy of APAR or GPP, and both of which do not change abruptly, to get the seasonal or spatial variation, the 4-day temporal resolution is adequate. Although higher temporal resolution may be obtained using the MODIS MCD43C4 dataset, larger spatial gaps caused by the low data quality (cloud, high aerosols) need to be filled. The information redundancy increases if excessive interpolation is applied. In addition, the MODIS MCD43C4 uses 16 days of input with the day of interest emphasized. This reduced the reliability of using MCD43C4 to represent the actual reflectance for the target date. Using 4-day temporal resolution will also decrease the dataset size compared to using a higher temporal resolution, and it is easy to aggregate and compare with other GPP dataset at 8-day or monthly temporal resolution.

Page 3, Lines 24-26: "In addition, OCO-2 can only generate a gridded monthly dataset at relatively coarse spatial resolution, typically at1°×1°, which limits its application in small regions. "

Reviewer: This is not quite correct. It is correct only if we limit our approach to bin- ning/averaging, and ignore spatial autocorrelations. However, if we take autocorre- lations into account, we get have estimates at much higher spatio-temporal resolu- tions (see Tadic et al, 2015, doi:10.5194/gmd-8-

3311-2015âA˘ land Tadic et al, 2017, doi:10.5194/gmd-10-709-2017)

Response: We agree that using geo-statistical methods, we can get higher spatial-temporal dataset from low resolution dataset. These methods are effective when the spatial autocorrelations are high (e.g., for atmospheric gases or atmospheric temperature). However, the surface vegetation heterogeneity is high especially in the presence of land use and land cover changes, making this method less applicable. Considering the large gaps between OCO-2 swaths (~100 km), using statistical method without additional information to generate high resolution SIF dataset from OCO-2 SIF would suffer from high uncertainty. Nevertheless, we revised this sentence to make it accurate, we also discussed the challenges of using statistical method to downscale the OCO-2 SIF dataset and cite the references suggested by the reviewer.

*"Although several statistical methods are proposed to downscale satellite observations to finer spatial-temporal resolutions (Tadić et al., 2015, 2017), considering the land surface heterogeneity and wide gaps between OCO-2 swaths (~ 100 km), it could be challenging to apply these methods to OCO-2 SIF."*

We also discussed the possibility of using geostatistical method to generate spatially contiguous drought monitoring dataset in Section 4.2:

*"The spatial coverage issues can be further improved using the geostatistical based method (Tadić et al., 2017), but this may need further investigation."*

Page 4, Line 17:" In this study, we aim to generate a global continuous SIF (CSIF) product. . ."

Reviewer: Here and throughout the text, perhaps better choice of word would be con- tiguous. Continuous implies that there is an infinite number of estimation locations, while in practice your estimation interval is determined by the granularity of input data, in this case MODIS retrievals.

Response: Thanks for your suggestion, we use the word continuous to distinguish from the swath based, 16-day revisit cycle OCO-2 SIF data. We agree that "contiguous" is more precise and we have revised the title and other occurrences throughout the manuscript.

Page 4, Line 32:" The reasons for using this resolution include: (1) it is directly compa- rable to the OCO-2 SIF footprint s

Reviewer: This statememnt is questionable, 5x5 gives 25km2 footprint, and OCO-2 footprint is less than 3km2 in size. 8 times difference might or might not be viewed as significant.

Response: We agree that, in terms of the area, 8 times difference can be regarded as significant. However, compared to other OCO-2 SIF aggregations (to 1 degree or 2 degree), they are still in the same order of magnitude. And OCO-2 retrievals can be much more representative for the gridcells SIF values of 0.05*0.05 than 1*1 or 2*2 degree. If we reduce the pixel sizes, we may get very few observations within each pixel and would not effectively reduce the uncertainty. We clarified this statement and have rewritten it as:

*"it is directly comparable (in the same order of magnitude) to the OCO-2 SIF footprint size (around 1.3km×2.25km) and the samples within each gridcell can be more evenly distributed and, thus, more representative of the gridcell SIF values than using much coarser $1° × 1°$ or $2° × 2°$ grids"*

Page 5, Lines 3-4:" assuming independent estimates and homogeneous SIF value within each gridcell. . ."

Reviewer: Here an additional assumption is required - the SIF has to be not only homogeneous (spatial dimension) but also constant in time.

Response: Since the aggregation is conducted for each day, that is, all the OCO-2 SIF retrievals used to calculate the average of the 0.05 by 0.05 gridcell are obtained with in a very short period of time (usually within several seconds). As long as we assume that SIF is homogeneous, this aggregation process can be regarded as multiple measurements of the same SIF source.

Page 6, Line 1:" For prediction, we first aggregated the daily reflectance to 4 days. " Reviewer:Why 4?

Response: We chose 4-day temporal resolution for the CSIF dataset as a balance between application requirements, information redundancy, and dataset sizes. We refer the reviewer to the previous responses for details.

We added a brief explanation why 4-day temporal resolution is used in this study.

Page 6, Lines 10-11:" A feedforward neural network (NN) is a number of computational nodes (called neurons) structured in a multi-layer architecture."

Reviewer:In principle, NN can be a single layer structure.

Response: Thanks for pointing this out, we have revised this sentence to

"A feedforward neural network (NN) is a number of computational nodes (called neurons) structured in a single or multi-layer architecture"

Page 6, Line 16:" The rectified linear unit (ReLU) was used as the activation " Reviewer:Is there any particular reason for this choice?

Response: The ReLU is frequently used as activation function for scientific research, although in computer sciences, sigmoid functions are also used for classification problems. Overall, ReLU has shown better performance than sigmoid functions in our application but also in most other ones. We
now clarify this point further.

Page 7, Lines 14-15:" RSIFGOME-2 (Gentine and Alemohammad, 2018a) uses a similar machine learning technique approach to CSIF but the 15 training is based on the bi-weekly gridded SIF product from GOME-2, and 8-day MYD09A1 reflectance dataset. "

Reviewer: This choice is surprising, as GOME-2 Level 3 products cn be obtained at much higher temporal resolutions, even daily, like it was demonstrated at Tadic et al., 2017. In this case, an unnecessary degradation of information content is induced, as temporal SIF variations during biweekly periods are converted into noise. Given large footprint on GOME-2 retrievals, ML processing here played the role of the downscaling as well, which itself is a challenging process.

Response: We believe that the RSIF dataset has its merits of reducing the uncertainties in raw GOME-2 SIF dataset and downscale GOME-2 SIF to higher spatial resolution. We agree that during this process, the within month variation are partly regarded as the noises, but this has limited effects since the GOME-2 SIF has relative high uncertainties for each individual observation. Since this was performed in another study we think it is beyond the scope of the current manuscript, but we clarify the point
suggested by the reviewer.

Page 8, Lines 1-6

Reviewer: More details are needed here, for the sake of reproducibility. Did you use regularization?
What kind of regularization? L2, dropout, their parametrizatuon? How many epochs? Did you use
test/validation sets approach or only test?

Response: We did not use dropouts nor other regularization method since the network was not very
deep and there was no sign of overfitting. We used 50 epochs with a batch size of 1024 for the
training. Only test approach is used. These are now clarified in the text.

Page 10. Line 8:" Figure 10 shows the difference between instantaneous clear-day OCO-2 SIF and
CSIFclear-inst. "

Reviewer: Using contiguous Level 3 products based on OCO-2 data and spatio- temporal kriging would
yield a figure equivalent to Fig 10, but contiguous.

Response: We thank the reviewer for the suggestion, as we have discussed previously, the application
of spatio-temporal kriging to OCO-2 SIF may be challenging and beyond the scope of this study. Here
we are only exploring the possibility of using the difference between SIF and CSIF to detect drought.
Nevertheless, it could be a very interesting study for the future to map drought impact contiguously
using the spatial-temporal kriging. We now acknowledge this in the text.